# Nuclear RPSA senses viral nucleic acids to promote the innate inflammatory response

Yan Jiang[1], Siqi Sun[1], Yuan Quan[1], Xin Wang[1], Yuling You[1], Xiao Zhang[1], Yue Zhang[1], Yin Liu[1], Bingjing Wang[1], Henan Xu[2] ✉ & Xuetao Cao [1,2] ✉

Innate sensors initiate the production of type I interferons (IFN-I) and proinflammatory cytokines to protect host from viral infection. Several innate nuclear sensors that mainly induce IFN-I production have been identified. Whether there exist innate nuclear sensors that mainly induce proinflammatory cytokine production remains to be determined. By functional screening, we identify 40 S ribosomal protein SA (RPSA) as a nuclear protein that recognizes viral nucleic acids and predominantly promotes proinflammatory cytokine gene expression in antiviral innate immunity. Myeloid-specific *Rpsa*-deficient mice exhibit less innate inflammatory response against infection with Herpes simplex virus-1 (HSV-1) and Influenza A virus (IAV), the viruses replicating in nucleus. Mechanistically, nucleus-localized RPSA is phosphorylated at Tyr204 upon infection, then recruits ISWI complex catalytic subunit SMARCA5 to increase chromatin accessibility of NF-κB to target gene promotors without affecting innate signaling. Our results add mechanistic insights to an intra-nuclear way of initiating proinflammatory cytokine expression in antiviral innate defense.

Host cells express various innate sensors for recognizing viral nucleic acids to trigger innate signaling cascades[1,2]. Several innate sensors in the cytoplasm have been reported to activate innate immune responses by jointly initiating the expression of type I interferons (IFN-I) and proinflammatory cytokines once recognizing viral nucleic acids. These cytoplasmic innate sensors include Cyclic GMP–AMP Synthase (cGAS)[3], Toll-like receptor-9 (TLR-9)[4], Z-DNA binding protein 1 (ZBP1)[5], Interferon-gamma inducible protein 16 (IFI16)[6], Interferon induced with helicase C domain 1 (MDA5)[1] and retinoic acid-inducible gene I (RIG-I)[7]. Some nuclear acids sensors, such as cGAS and RIG-I, have also been found to be translocated to the nucleus and initiate innate immune responses against virus infection[8–10]. In a previous study, we reported the heterogeneous nuclear ribonucleoprotein A2/B1 (hnRNPA2B1) as a nuclear DNA sensor for the induction of IFN-I in response to DNA virus infections[11]. Nevertheless, whether and how viral nucleic acids are recognized in the nucleus to selectively induce the transcription of proinflammatory cytokine genes for enhancing antiviral innate response is still largely unknown.

Inflammation is indispensable for the host to resist viral infection. Once recognizing the invading virus, the innate immune systems can initiate either inflammatory signaling pathways such as NF-κB or inflammasome responses such as NLRP3 and AIM2 for activating antiviral inflammatory responses[12,13]. In acute hepatitis B virus (HBV) infection, the DNA virus usually inadequately induces expression of type I/III interferons and ISGs but induces expression of proinflammatory cytokines[14], which have been reported to inhibit HBV infection in an interferon-independent manner and reduce host susceptivity[15–17]. Moreover, the proinflammatory cytokines are critical in clearing viral infections by inducing effector lymphocyte activation and recruiting neutrophils[18]. Efficient launching of inflammatory response is essential for the host anti-virus process. DNA viruses, such as herpes simplex virus type 1 (HSV-1), mainly release their genomes

[1]Department of Immunology, Center for Immunotherapy, Institute of Basic Medical Sciences, Peking Union Medical College, Chinese Academy of Medical Sciences, Beijing 100005, China. [2]Frontiers Science Center for Cell Responses, Institute of Immunology, College of Life Sciences, Nankai University, Tianjin 300071, China. ✉e-mail: xuhenan_immuno@nankai.edu.cn; caoxt@immunol.org

from their capsids directly into the nucleus after invading host cells[19], which can inhibit the activation of innate signaling pathways in the cytoplasm to escape from host immune discrimination[20,21]. Viral DNA in the nucleus can also hijack host histones to form nucleosome structure, subsequently keeping the host in immune silenced status[22]. RNA viruses like Influenza A virus (IAV) also replicate in the host nucleus. Identification and a more comprehensive investigation of innate sensors in the cell nucleus, previously known or unknown, for launching innate immune responses is essential for a better understanding of how host cells defend against viral infection and also helpful in designing therapeutic strategies for infectious diseases.

The multifunction molecule 40 S ribosomal Protein SA (RPSA) is implied to be an HSV-1 DNA binding protein candidate in the host nucleus[11]. Here, by functional screening, we report the identification of RPSA as a nuclear innate sensor that selectively induces the transcription of proinflammatory cytokines genes, which is dependent on NF-κB signaling. We show that, upon sensing of nucleic acids from HSV-1 or IAV, nucleus-localized RPSA is selectively responsible for starting epigenetic modification reconstruction and enhancing P65 subunit enrichment at proinflammatory cytokine gene promoters but not affecting NF-κB signaling cascades in launching antiviral innate responses. Our study sheds light on the understanding of host-virus interaction by unveiling a new nuclear innate sensor RPSA which selectively promotes NF-κB-triggered proinflammatory cytokine expression through modulating epigenetic modification. Our results also provide a potential target for the control of viral infectious diseases.

## Results

### Nuclear RPSA is a viral nucleic acids-binding molecule

To explore the potency of the 23 viral DNA-binding nuclear protein candidates we obtained previously[11] in regulating proinflammatory cytokine expression, we used HSV-1-infected mouse peritoneal macrophages (PMs) as a model to perform the functional screening with a small RNA interference library. The screening revealed that knocking down *Rpsa* robustly repressed HSV-1-induced expression of *Il1b*, the typical anti-viral cytokine, as well as a set of proinflammatory cytokines genes, suggesting RPSA as an important molecule in virus infection-triggered innate inflammatory response (Fig. 1a and Supplementary Fig. 1a, b). Notably, knocking down *Rpsa* had no impact on the intracellular level of HSV-1 DNA and the gene expression of *Ifnb* or *ISGs*, such as *Ifit1* and *Rsad2* (Supplementary Fig. 1c). Broadly, in tetracycline-induced *Rpsa* deficient RAW264.7 cells (Supplementary Fig. 1d), proinflammatory cytokine gene expression was robustly decreased due to *Rpsa* depletion in both HSV-1 and IAV infection (Supplementary Fig. 1e, f). In human A549 epithelial cells, knocking out *RPSA* impaired the proinflammatory cytokine and chemokine gene expression in HSV-1 or IAV infection without reducing the *IFNB* expression and the virus levels (Fig. 1b, c). Co-immunoprecipitation assays using an RPSA-specific antibody and the nuclear lysate from viral-infected mouse bone marrow-derived macrophages (BMDMs) showed that RPSA was capable of interacting with both HSV-1 DNA and IAV genome RNA in infection (Fig. 1d, e). Strikingly, the immunofluorescence assay exhibited that RPSA co-localized with EdU-labeled HSV-1 genome DNA or the EU-labeled IAV genome RNA (gRNA) within the host nucleus (Fig. 1f, g). Consistent, the in situ association of RPSA with EdU-labeled HSV-1 genome DNA was observed during infection in DNA pull-down assay (Fig. 1h). The interaction of nuclear RPSA with biotinylated DNA could be blocked by the unlabeled DNA (Fig. 1i). These data suggested that nuclear located RPSA bound viral nucleic acids during infection.

RPSA direct binding to HSV-1 DNA was further proved by immunoprecipitation assay using recombinant mouse RPSA (rmRPSA) and biotin-labeled HSV-1 DNA, and their binding could be blocked by the unlabeled HSV-1 DNA in a dose-dependent manner. The mouse naked genome DNA, but not nucleosomes, also competitively blocked RPSA

binding to biotin-HSV-1 DNA (Fig. 1j). EMSA revealed that the binding of RPSA to the biotin-HSV-60 probe could be competitively blocked by unlabeled HSV-60 itself as well as by VACV-70 dsDNA, poly dA:dT and poly dI:dC (Supplementary Fig. 1g). Mammalian RPSA is sequence conserved. Recombinant human RPSA (rhRPSA) protein also bound HSV-60 dsDNA or IAV gRNA, which was blocked by different sources of DNA or the free IAV gRNA, respectively (Fig. 1k, l). RPSA consists of 295 amino acids (a.a), and its N-terminus, a.a. 1–209, is mainly involved in the composition of the ribosome structure. The C-terminus of RPSA, a.a. 210–295, is primarily engaged in laminin properties and interactions with other proteins or rRNA. Pull-down experiments showed that only the full-length RPSA interacted with HSV-1 DNA (Fig. 1m).

Thus, our data demonstrated that RPSA directly binds viral DNA and RNA in the nucleus during HSV-1 and IAV infection.

### RPSA accelerates proinflammatory cytokine expression upon recognizing viral nucleic acids within the nucleus without affecting IFN-β production

To clarify the function of RPSA in the innate response against viral infection, we disturbed the endogenous *Rpsa* expression in multiple cell types. The myeloid-specific *Rpsa* deficient mice were generated (Supplementary Fig. 2a–c). Conditional loss of *Rpsa* did not affect the development of immune cells, as the percentages of F4/80⁺CD11b⁺ macrophages, granulocytes, dendritic cells were comparable in the spleens of *Rpsa*^fl/fl^Lyz-Cre⁺ mice and *Rpsa*^fl/fl^ littermates (Supplementary Fig. 2d). Upon HSV-1 challenge, BMDMs from *Rpsa*^fl/fl^Lyz-Cre⁺ mice expressed lower levels of *Il1a*, *Il1b*, *Il6*, *Tnfa*, and *Il12b* mRNAs and secreted lower levels of corresponding cytokine proteins. Notably, loss of *Rpsa* rarely affected the expression of *Ifnb* (Fig. 2a, b).

Overexpression of RPSA upregulated proinflammatory cytokine gene expression in MEFs upon HSV-1 infection (Fig. 2c). Similarly, HSV-1-induced proinflammatory cytokine expression also decreased significantly in *Rpsa* deficient murine lung epithelial-12 (MLE-12) cells, which could be restored by ectopic expression of RPSA (Fig. 2d, Supplementary Fig. 3a). The results indicated a broad role of RPSA in promoting proinflammatory cytokine expression against HSV-1 infection in various cell types. Correspondingly, the unbiased RNA-seq analysis of tetracycline-induced *Rpsa* deficient RAW264.7 cells, with or without HSV-1 infection for 4 h, revealed that the loss of *Rpsa* impaired the expression of more than 1500 genes (Supplementary Fig. 3b, c), which were enriched in pathways related to defense response to the virus and inflammatory response indicated by KEGG enrichment analysis (Supplementary Fig. 3d). Figure 2e shows the heatmap of immune-related genes whose expression was downregulated after HSV-1 infection due to *Rpsa* deletion. These results proved that RPSA selectively promotes the expression of proinflammatory cytokine genes but not IFN-I during infection with HSV-1 in the nucleus.

Importantly, nucleofection of naked dsDNA or dsRNA activated *Il1b* and *Il6* mRNA expressions in RAW264.7 cells, which were reduced by *Rpsa* deficiency (Fig. 2f, g). However, *Rpsa* deficiency had little effect on proinflammatory cytokine expression upon cytoplasmic transfection of either TLR9 ligand CpG oligonucleotide (Supplementary Fig. 3e), poly dA:dT (Fig. 2h), TLR3/RIG-I ligand poly I:C (Fig. 2i). Also, the loss of or overexpression of RPSA did not affect the proinflammatory cytokines gene expression upon TNF-α or cGAMP stimulation (Supplementary Fig. 3f, g). Taken together, the results further confirmed that nuclear RPSA promotes the expression of proinflammatory cytokines by sensing viral nucleic acids within the nucleus.

### RPSA preferentially increases chromatin accessibility of proinflammatory cytokine gene promotors after HSV-1 infection

Viral entry of host cells can immediately trigger proinflammatory gene expression by the activation of NF-κB and MAPK signaling cascades mainly depending on TAK1 mediated TLR pathways and cGAS-STING axis. However, both pathways in WT and *Rpsa* deficient RAW264.7 cells

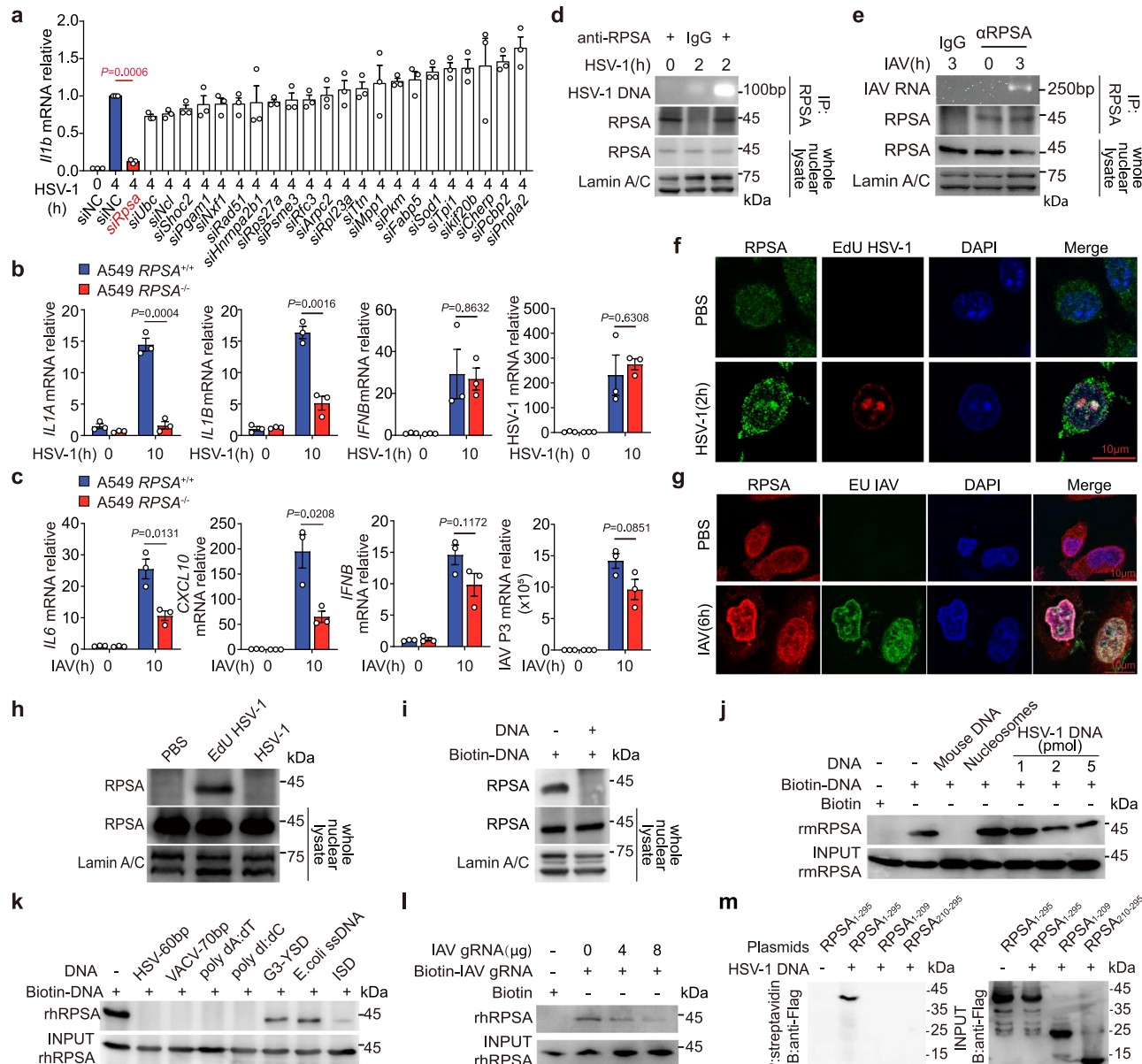

**Fig. 1 | RPSA binds to viral nucleic acids and activates proinflammatory cytokine expression in macrophages. a** qRT-PCR analysis of *Il1b* mRNA in peritoneal macrophages (PMs) infected with HSV-1 for 8 h after interfering with the indicated genes (*n* = 3). **b, c** Wild-type and *RPSA*-KO A549 epithelial cells were infected with HSV-1(MOI,10) (**b**) or IAV (MOI, 1) (**c**) for 10 h, then qRT-PCR determined levels of indicated mRNAs (*n* = 3). **d** PCR analysis of HSV-1 DNA in the complex immunoprecipitated by an anti-RPSA antibody or IgG from the nucleus of bone marrow-derived macrophages (BMDMs) infected with HSV-1. **e** RT-PCR analysis of IAV gRNA immunoprecipitated by an anti-RPSA antibody from IAV-infected BMDMs nucleus. **f, g** Co-localization of RPSA (green) and EdU-labeled HSV-1 genome (red) (**f**) in BMDMs or RPSA (red) and EU-labeled IAV genome (green) in A549 (**g**) without or with infection for indicated times were examined by confocal microscopy. Nuclei were stained with DAPI (blue). Scale bar = 10 μm. **h** Proteins pulled down from the nucleus of BMDMs infected with unlabeled or EdU-labeled HSV-1 for 2 h were analyzed by immunoblot. **i** Nuclear complexes obtained by nuclear acid affinity purification were examined by immunoblot in the absence or presence of unlabeled HSV-1 DNA. **j** Recombinant mouse RPSA (rmRPSA) was incubated with biotin-labeled HSV-1 DNA without or with mouse naked genome DNA or nucleosomes, or increasing concentrations of unlabeled HSV-1 DNA, and pull-downed RPSA was detected by immunoblot. **k** Recombinant human RPSA (rhRPSA) was incubated with biotin-labeled HSV-60 dsDNA without or with the indicated unlabeled nucleic acids, and then the pull-downed RPSA was detected by immunoblot. **l** rhRPSA was incubated with biotin-labeled IAV gRNA without or with increasing concentrations of unlabeled IAV gRNA, nuclear acid was affinity purified, and RPSA was detected by immunoblot. **m** Flag-tagged full-length or truncated RPSA proteins were incubated with biotinylated HSV-1 DNA and DNA-bound proteins were examined by immunoblot. Similar results were obtained for three independent experiments. One representative experiment is shown. Data in **a–c** are shown as mean ± s.e.m. The *P* values were calculated by a two-tailed unpaired Student's *t*-test. Source data are provided as a Source Data file.

were comparable in response to HSV-1 infection (Supplementary Fig. 4a). Moreover, loss of *Rpsa* neither affect the nuclear translocation of phosphorylated P65 (Supplementary Fig. 4b, c), P42/44, P38 and IRF3 (Supplementary Fig. 4b), nor affect apoptosis, pyroptosis or necroptosis upon HSV-1 infection (Supplementary Fig. 4d). Further investigation showed that the enhanced *Il1b* and *Il6* expressions by

RPSA overexpression was vanished in MLE-12 cells once treated with a combined inhibitor of TAK1 and STING (pan-inhibitor). Notably, the overexpression of RPSA could restore the proinflammatory cytokine gene expression when the inhibitor blocked one single pathway (Fig. 3a). Consistently, inhibition of STING in RPSA knocking downed PMs further reduced *Il1b* and *Il12b* expression in HSV-1 infection

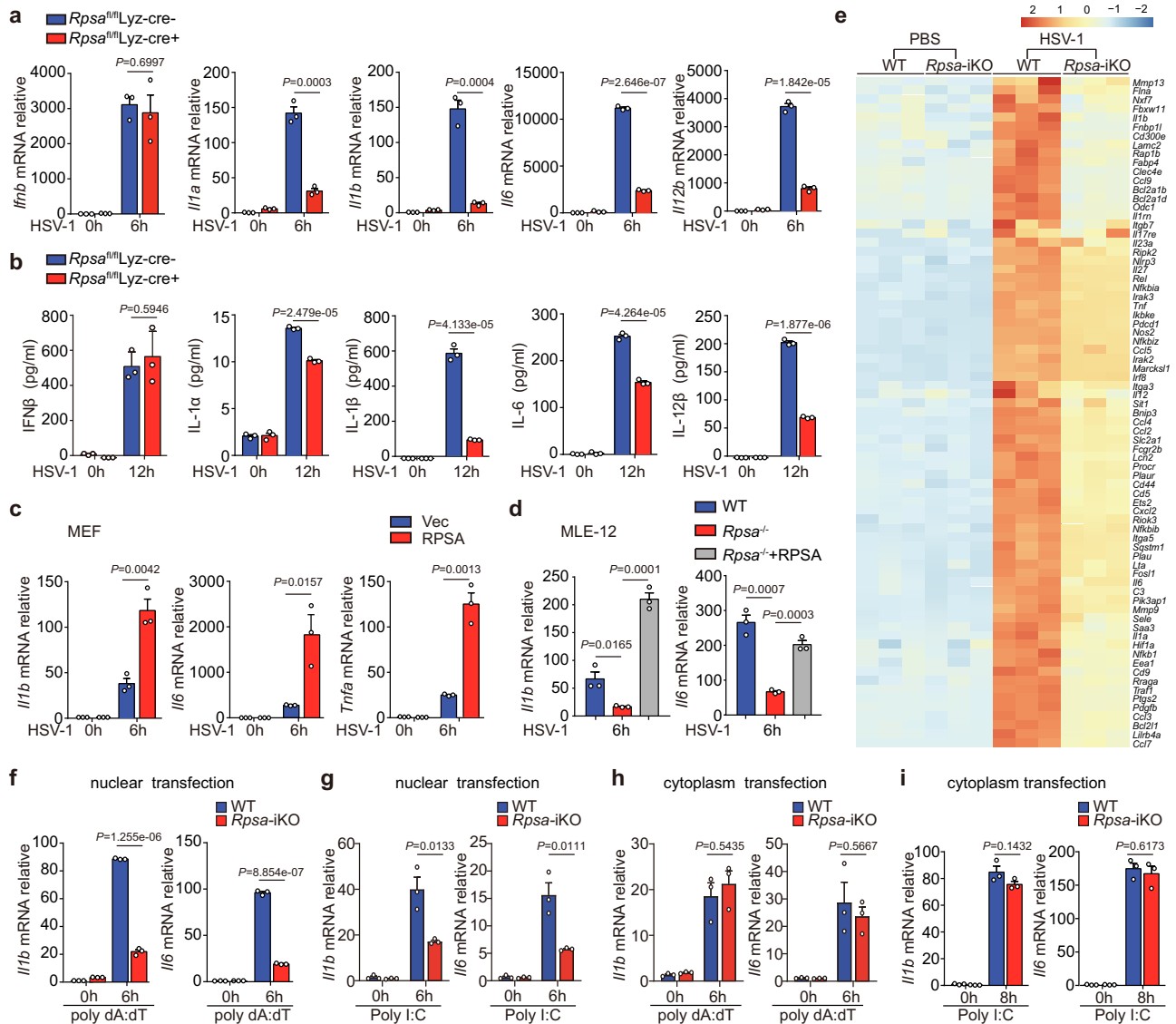

**Fig. 2 | Nuclear RPSA senses viral nucleic acids and accelerates proinflammatory cytokine expression upon HSV-1 infection. a, b** BMDMs derived from *Rpsa*fl/fl Lyz-Cre+ mice and the littermates were infected with HSV-1 for the indicated times and then analyzed for *Ifnb*, *Il1a*, *Il1b*, *Il6*, and *Il12b* mRNAs by qRT-PCR (**a**) (*n* = 3) and for protein levels in the cell culture supernatant by ELISA (**b**) (*n* = 3). **c** MEFs transfected with empty vector (Vec) or RPSA-expressing plasmid for 24 h were infected with HSV-1 then *Tnfa*, *Il1b*, and *Il6* mRNA levels were examined by qRT-PCR (*n* = 3). **d** Wild-type and *Rpsa*-iKO MLE-12 cells were reconstituted with RPSA-expressing plasmid followed by qRT-PCR analysis of *Il1b* and *Il6* mRNAs after HSV-1 infection for 6 h (*n* = 3). **e** Heatmap of downregulated immune and inflammatory

response genes in *Rpsa*-iKO RAW264.7 cells relative to in wild-type RAW264.7 cells in response to HSV-1 infection. **f, g** Wild-type and *Rpsa*-iKO RAW264.7 cells were nuclear transfected with poly dA:dT (**f**) or poly I:C (**g**), and *Il1b* and *Il6* mRNAs were analyzed by qRT-PCR 6 h later (*n* = 3). **h, i** qRT-PCR analysis of *Il1b and Il6* mRNAs in wild-type and *Rpsa*-iKO RAW264.7 cells with liposome dA:dT transfection (**h**) or poly I:C (**i**) for the indicated times (*n* = 3). Similar results were obtained from three independent experiments and one representative experiment is shown (**a–d** and **f–i**). Data in **a–d**, **f–i** are shown as mean ± s.e.m. The *P* values were calculated by a two-tailed unpaired Student's *t*-test. Source data are provided as a Source Data file.

(Supplementary Fig. 5a). Collectively, the results indicated that RPSA enhancement of proinflammatory cytokine gene expression depends on and makes up for the innate signaling cascade activation without affecting the signal transduction.

Digital genomic footprinting of transposase accessible chromatin with high throughput sequencing (ATAC-seq) assay exhibited reduced peak enrichment levels around the transcription start site (TSS) in *Rpsa*-iKO RAW264.7 cells than that in WT cells infected with HSV-1 (Supplementary Fig. 5b). Figure 3b showed the clustering results of differential ATAC-peaks in WT and *Rpsa*-deficient RAW264.7 cells after HSV-1 infection. The line graph confirmed a reduced ATAC-seq signal in the promoter regions of *Il1b* but not the *Ifnb* gene (Fig. 3c). KEGG functional enrichment analysis showed that differential

peak-associated genes whose enrichment levels were reduced by *Rpsa* deletion mainly concentrated in innate immunity and inflammation-related diseases as well as downstream genes in signaling pathways of NF-κB, TLR and TNF (Supplementary Fig. 5c). The DNase I hypersensitivity assay further verified the ATAC-seq data. The loss of *Rpsa* significantly reduced the chromatin accessibility at *Il1b*, *Il6*, *Il12b* and *Tnfa* promoter regions but did not affect the *Ifnb* promoter in RAW264.7 cells upon HSV-1 infection (Fig. 3d). We also checked chromatin accessibility in HSV-1 infected *Rpsa*-iKO RAW264.7 cells pretreated with or without STING inhibitor (Supplementary Fig. 5d). The results indicated that the inhibitor treatment showed little effect on the DNase sensitivity of *Il1b*, *Il12b* and *Tnfa* promoter regions. Accompanying the DNase I accessibility assays in cGAMP-stimulated *Rpsa*-iKO

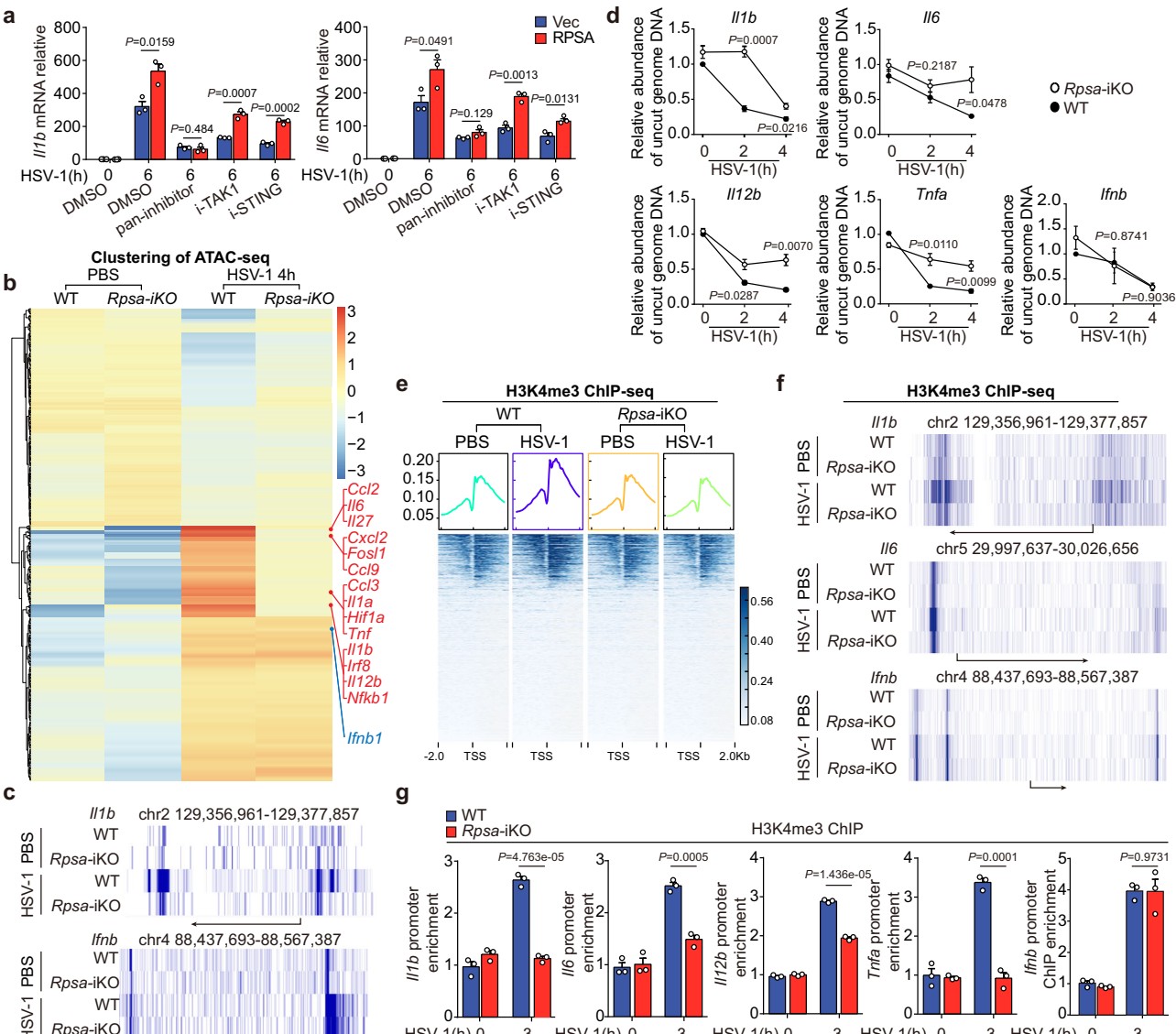

**Fig. 3 | RPSA facilitates virus infection-induced proinflammatory cytokine expression via enhancing chromatin accessibility. a** qRT-PCR analysis for *Il1b* and *Il6* mRNA levels in MLE-12 cells transfected with the indicated plasmids. Then pre-treated with pan-inhibitor (Takinib 10 μM & C-178 5 μM), TAK1 inhibitor (Takinib 10 μM), STING inhibitor (C-178 5 μM) or DMSO for 2 h and infected with HSV-1 for 6 h (*n* = 3). **b-c** ATAC-seq assay of wild-type and *Rpsa*-iKO RAW264.7 cells infected with or without HSV-1 infection. Heatmap of clustering of genes associated with differential peaks (**b**) and seq-signal in the transcribed region of *Il1b* and *Ifnb* (**c**) are presented. **d** DNase I sensitivity assay at promoter regions of *Il1b*, *Il6*, *Il12b*, *Tnfa*, and *Ifnb* in wild-type and *Rpsa*-iKO RAW264.7 cells infected with HSV-1 for the indicated time (*n* = 3). **e, f** H3K4me3 ChIP-seq assay of wild-type and *Rpsa*-iKO RAW264.7 cells with or without HSV-1 infection. Distribution of reads relative to TSS (**e**) and seq-signal in the transcribed region of *Il1b*, *Il6*, and *Ifnb* (**f**) are presented. **g** ChIP-qPCR assay of H3K4me3 at *Il1b*, *Il6*, *Il12b*, *Tnfa* and *Ifnb* promoter sites in wild-type and *Rpsa*-iKO RAW264.7 cells (*n* = 3). Similar results were obtained from three independent experiments and one representative experiment is shown (**a**, **d**, and **g**). Data in **a**, **d**, **g** are shown as mean ± s.e.m. The *P* values were calculated by a two-tailed unpaired Student's *t*-test. Source data are provided as a Source Data file.

and the wild-type controls (Supplementary Fig. 5e), our data showed that RPSA promoted chromosome accessibility independent of STING activation. Thus, RPSA promoted HSV-1-induced expression of proinflammatory cytokine genes by enhancing chromatin accessibility of the promoter regions.

The tri-methylation of histone H3 at lysine 4 (H3K4me3) around the TSS of target genes was the hallmark of the effectively discriminating genes with high transcriptional activation[23]. Analysis of H3K4me3 ChIP-seq data from HSV-1 infected WT and *Rpsa*-deficient RAW264.7 cells showed that 20% of the differential peaks were distributed in promoter regions (Supplementary Fig. 5g). The heat map showing the enrichment of H3K4me3 around the TSS of WT and *Rpsa*-deficient cells revealed that the loss of *Rpsa* reduced the enrichment,

suggesting decreased transcriptional activity (Fig. 3e). The line graph confirmed reduced H3K4me3 levels in the promoter regions of *Il1b*, and *Il6*, further supporting that *Rpsa* deficiency reduced the transcription of proinflammatory cytokine genes but not *Ifnb* genes (Fig. 3f). The KEGG pathway analysis of differential peaks showed that the loss of RPSA reduced the expression of many genes involved in inflammatory diseases like rheumatoid arthritis and Alzheimer's disease (Supplementary Fig. 5f). The significantly reduced enrichment of H3K4me3 in the promoter regions of *Il1b*, *Il12b*, *Tnfa* and *Il6*, rather than *Ifnb*, in the absence of *Rpsa* was also confirmed by the ChIP-qPCR assay (Fig. 3g).

Taken together, our data have demonstrated that RPSA, sensing nucleic acids in the host nucleus, then selectively promotes the

proinflammatory cytokine gene transcription through epigenetically remolding chromosome accessibility in response to viral infection.

## RPSA enhances proinflammatory cytokine gene transcription by interacting with SMARCA5

To further elucidate the underlying mechanism, we performed co-immunoprecipitation-mass spectrometry (CoIP-MS) analysis to identify RPSA-interacting proteins. Using unlabeled quantitative proteomics analysis, we found that 316 proteins associated with RPSA were up-regulated and 621 were down-regulated in macrophages upon HSV-1 infection (Supplementary Fig. 6a). Protein domains with increased interaction with RPSA included bromodomains, nuclear SNF2-related domains, and other domains associated with chromatin remodeling proteins (Supplementary Fig. 6b), suggesting that RPSA might be involved in the interaction with chromatin remodeling complexes. By integrating the mass spectrometry analysis of differential bands by CoIP-MS and quantitative proteomics results, the top 10 protein candidates overlapped. The SWI/SNF-related matrix-associated actin-dependent regulator of chromatin subfamily A member 5 (SMARCA5) was identified as the most substantial RPSA-interacting protein candidate with the highest score and coverage percentage (Supplementary Fig. 6c, d), which was further confirmed by nuclear co-immunoprecipitation (Fig. 4a). In line with the previous phenotypes, we detected RPSA/SMARCA5 interacted with NF-κB P65 subunit after HSV-1 infection, while IRF3 was excluded from the complex, further indicating *Ifnb* was not a target gene of RPSA/SMARCA5 axis (Fig. 4a). Consistently, the interaction of RPSA with SMARCA5 was also increased in macrophages upon IAV infection (Fig. 4b).

SMARCA5 is the core catalytic unit of the ISWI (Imitation Switch) chromatin-remodeling complex that facilitates the transcriptional initiation of immune molecules, including proinflammatory cytokines[24]. ChIP analysis showed that SMARCA5 was enriched at the promoter regions of *Il1b*, *Il6* and *Il12b* in macrophages after HSV-1 infection, which was significantly reduced in *Rpsa* deficient macrophages (Fig. 4c). In HSV-1-infected MLE-12 cells, higher *Il1b* and *Il6* expressions rescued by SMARCA5 overexpression were vanished by deletion of RPSA. Nevertheless, SMARCA5 enhanced *Ifnb* expression independent of RPSA (Fig. 4d). Therefore, the ISWI chromatin remodeler SMARCA5 was required for RPSA-promoted transcriptional activation of proinflammatory cytokine genes.

## RPSA phosphorylation at Tyr204 is required for its interaction with SMARCA5

Post-translational modifications play important roles in the regulation of protein function. Tyrosine phosphorylation of RPSA increased significantly while serine/threonine phosphorylation and acetylation of RPSA did not change significantly after HSV-1 infection (Fig. 4e). The enhanced tyrosine phosphorylation showed independent of cGAMP stimulation (Supplementary Fig. 6e). More importantly, different from cytosolic RPSA, only nuclear RPSA showed an increase in tyrosine phosphorylation in HSV-1 and IAV infection within 4 h (Fig. 4f, g). Of note, the cytoplasm-replicated vaccine virus (VACV) infection could not enhance the tyrosine phosphorylation events (Supplementary Fig. 6g).

Mutation analysis of tyrosines within RPSA revealed that the mutation of Tyr204 to alanine (Y204A) abolished *Il1b* expression and tyrosine phosphorylation induced by HSV-1 infection, indicating that Tyr204 was vital for RPSA to promote the expression of proinflammatory cytokines after HSV-1 infection (Supplementary Fig. 6f and Fig. 4h). In contrast to wild-type RPSA, ectopic expressing RPSA-Y204A in *Rpsa*-deficient MLE-12 cells did not rescue the expression level of *Il1b* in HSV-1 or IAV infection. On the other hand, overexpression of RPSA-Y204D, mimicking persistent tyrosine phosphorylation, significantly up-regulated *Il1b* expression induced by HSV-1 and IAV infection.

Significantly, *Il1b* level was not affected by expressing RPSA-Y204A but was still up-regulated by expressing RPSA-Y204D after cytoplasmic CpG ODN stimulation, confirming nuclear RPSA was required for boosting inflammatory factor gene expression in conjunction with cytoplasmic immune signals (Fig. 4i).

The interaction of RPSA-Y204A with SMARCA5 was significantly reduced in HEK293T cells upon HSV-1 infection as compared with RPSA-WT, whereas RPSA-Y204D persistently interacted with SMARCA5 (Fig. 4j). Together, phosphorylation at Tyr204 is vital for RPSA to interact with SMARCA5 for promoting the expression of proinflammatory cytokines upon HSV-1 infection.

## RPSA facilitates P65 enrichment at proinflammatory cytokine gene promotors

As we aligned the motifs enriched by these differential peaks of H3K4 ChIP-seq data and found that the H3K4me3 level of NF-κB-binding motifs was reduced (Fig. 5a). Further investigation showed that the restored *Il1b* and *Il6* expressions by RPSA overexpression vanished in *Rpsa*-iKO MLE-12 cells once treatment with NF-κB inhibitor (BAY11-7082) (Fig. 5b), indicating that RPSA enhancement of *Il1b* and *Il6* expressions totally depends on NF-κB signaling cascade activation. We supposed that deficiency in *Rpsa* reduced interaction between chromatin remodeling complexes and NF-κB. The Co-IP result demonstrated that SMARCA5 binding to P65 was increased upon HSV-1 infection (Fig. 5c) but not upon VACV infection (Supplementary Fig. 7a). Strikingly, the interaction between SMARCA5 and P65 was indeed lost in *Rpsa* deficient cells (Fig. 5c). In further, the interaction between RPSA and P65 vanished when SMARCA5 was knockdown, suggesting that recognition of nucleic acids and interaction with ISWI chromatin remodeling complex is critical for RPSA to facilitate P65 binding to proinflammatory factor promoters (Fig. 5d).

The ChIP-seq results showed that the P65 enrichment around TSS significantly reduced in the absence of *Rpsa* (Fig. 5e). The clustering of differential enrichment peaks showed loss of *Rpsa* preferentially reduced P65 enrichment at the proinflammatory cytokine gene promoters rather than the interferons after HSV-1 infection (Supplementary Fig. 7b). KEGG pathway analysis showed that reduced enrichments of P65 with the loss of RPSA were associated with many signaling pathways related to the expression of proinflammatory cytokines, including the NF-κB signaling pathway and inflammatory diseases (Fig. 5f). We then performed ChIP-qPCR analyses of NF-κB P65 subunit. The *Rpsa* deficiency significantly reduced the enrichment levels of P65 in the promoter regions of *Il1b*, *Il6*, and *Il12* rather than *Ifnb* in RAW264.7 cells and BMDMs in HSV-1 infection (Fig. 5g, Supplementary Fig. 7c). We also performed ChIP-qPCR analyses of RNA polymerase II (Poly II). The enrichment levels of poly II in the promoter regions of *Il1b*, *Il6*, *Il12b*, and *Tnfa* in RAW264.7 cells and BMDMs in response to HSV-1 infection were significantly reduced in the absence of *Rpsa* (Fig. 5h, Supplementary Fig. 7d). Of note, enrichment of poly II at *Ifnb* promoter remained unchanged in *Rpsa*-deficient RAW264.7 cells upon infection (Fig. 5h). Taken together, decreased expression of proinflammatory cytokines due to *Rpsa* deletion is on account of reduced enrichment of P65 at proinflammatory cytokine gene promoters, rather than impairing P65 activation.

## RPSA is required for antiviral innate response in vivo

To evaluate the physiological importance of RPSA in the host defense against nuclear-replicating virus infection, we first infected *Rpsa*^fl/fl^Lyz-Cre^+^ mice and littermates with HSV-1. The viral load was significantly higher in the lung and brain (Fig. 6a), and *Il1b*, *Il6*, and *Tnfa* mRNA levels were significantly lower in the blood of *Rpsa*-deficient mice than those in littermates after HSV-1 infection (Fig. 6b). Accordingly, levels of IL-1β, IL6, and TNFα in sera were significantly lower in *Rpsa*-deficient mice. Still, the IFN-β levels were similar (Fig. 6c).

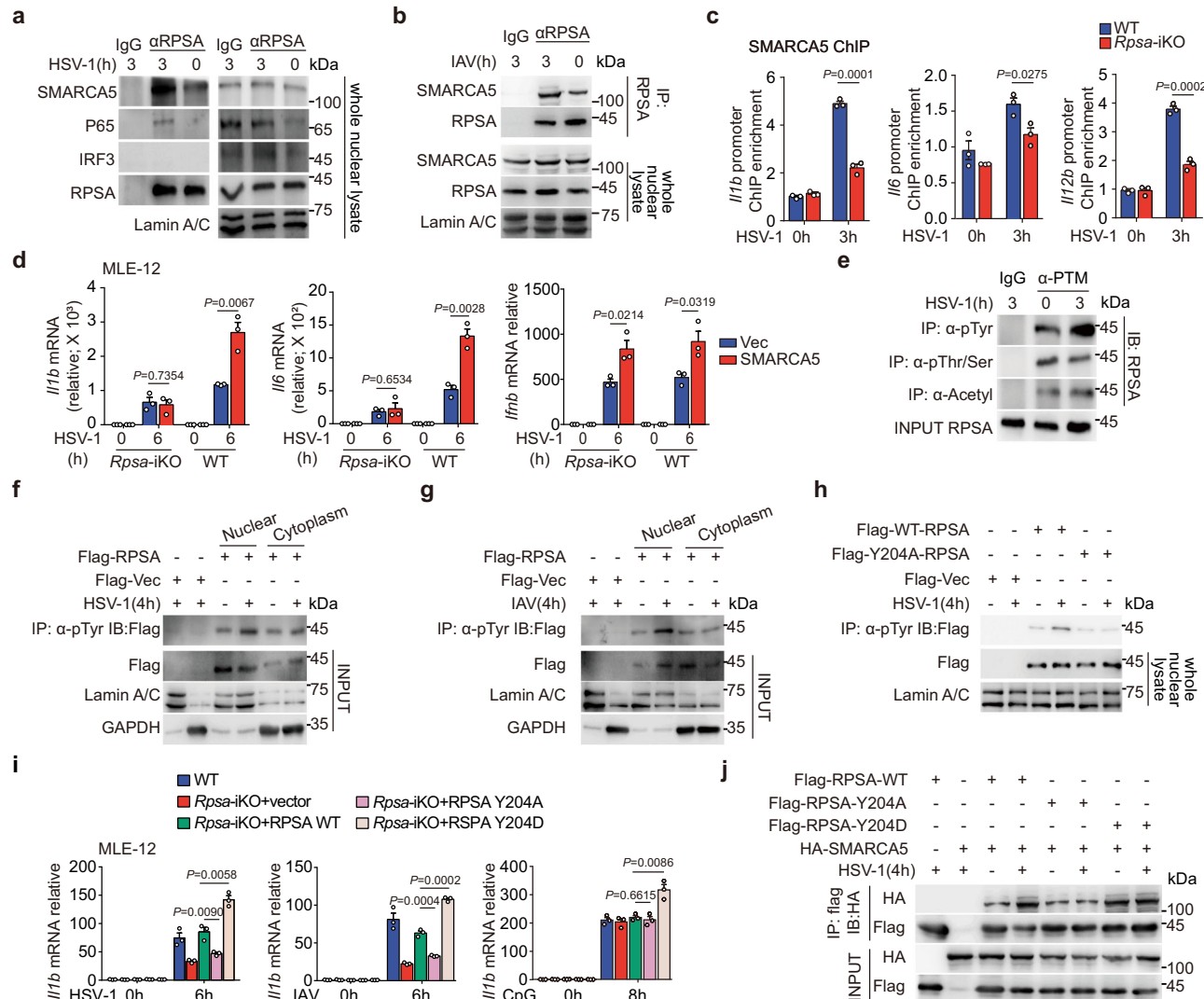

**Fig. 4 | RPSA promotes proinflammatory cytokine gene transcription upon viral infection by interacting with SMARCA5. a, b** Nuclear extracts from RAW264.7 cells infected with HSV-1 (**a**) or IAV (**b**) were immunoprecipitated with anti-RPSA antibody or IgG and then immunoblotted for indicated proteins. **c** ChIP-qPCR assay of SMARCA5 recruitment to the *Il1b*, *Il6*, and *Il12b* promoter sites in wild-type and *Rpsa*-iKO RAW264.7 cells infected with or without HSV-1 (*n* = 3). **d** *Il1b*, *Il6*, and *Ifnb* mRNA levels in wild-type and *Rpsa*-iKO MLE-12 cells transfected with the indicated plasmids then infected with HSV-1 were examined by qRT-PCR (*n* = 3). **e** The total proteins with specified posttranslational modifications were obtained by immunoprecipitation with indicated antibodies with or without HSV-1 infection. Then RPSA was detected through immunoblot analysis. **f, g** Cell lysates from HEK293T cells, which were transfected with the indicated plasmids and then infected with HSV-1 (**f**) or IAV (**g**), were immunoprecipitated with anti-phospho-

tyrosine antibody (α-Tyr) and then RPSA was detected through immunoblot analysis. **h** Subcellular compartment from HEK293T cells, which were transfected with the indicated plasmids and then infected with HSV-1, were immunoprecipitated with α-Tyr and then immunoblotted with indicated antibodies. **i** qRT-PCR analysis for *Il1b* mRNA levels in wild-type or *Rpsa*-iKO MLE-12 cells transfected with the indicated plasmids then infected with HSV-1, IAV, or stimulated with 5 μM CpG ODNs (*n* = 3). **j** Cell lysates from HEK293T cells transfected with the indicated plasmids and then infected with HSV-1 were immunoprecipitated with anti-Flag antibody and then immunoblotted with indicated antibodies. Similar results were obtained for three independent experiments. One representative experiment is shown. Data in **c, d, i** are shown as mean ± s.e.m.. The *P* values were calculated by a two-tailed unpaired Student's *t*-test. Source data are provided as a Source Data file.

Compared with littermates, the expression of *Il1b*, *Il6*, and *Il12b* in the brain (Fig. 6d) and the expression of *Il1b*, *Il6*, and *Tnfa* in the lung (Fig. 6e) and liver (Fig. 6f) of *Rpsa*-deficient mice were significantly decreased. The hematoxylin-eosin (HE) staining showed lower severity of inflammation in the lung and liver (Fig. 6g, and Supplementary Fig. 8a). Consistently, immunohistochemical analysis of the lung of *Rpsa*-deficient mice showed that the levels of IL-1β (Fig. 6h) and IL-6 (Fig. 6i) were reduced. Thus, RPSA is required for the efficient induction of the innate inflammatory response against HSV-1 infection in vivo.

Moreover, we infected *Rpsa*^fl/fl^Lyz-Cre^+^ mice and littermates with IAV. The IAV load was significantly increased in the lung (Fig. 6j) and *Rpsa*^fl/fl^Lyz-Cre^+^ mice also showed reduced inflammatory responses to

IAV infection with comparable levels of IFN-β (Fig. 6k, l). Taken together, nuclear sensor RPSA is essential for host defense against nuclear-replicating virus infection in vivo.

## Discussion

How host cells accurately recognize viral nucleic acids and initiate innate responses to defend against viral infection remains to be fully understood. Here, we identified nuclear RPSA as an innate sensor that promotes the host's innate inflammatory response against virus infection (Supplementary Fig. 8b). This study improves our understanding of the innate inflammatory responses that start from the host nucleus. Also, it suggests a complex regulatory mechanism in the nucleus to help the host respond to viral infection.

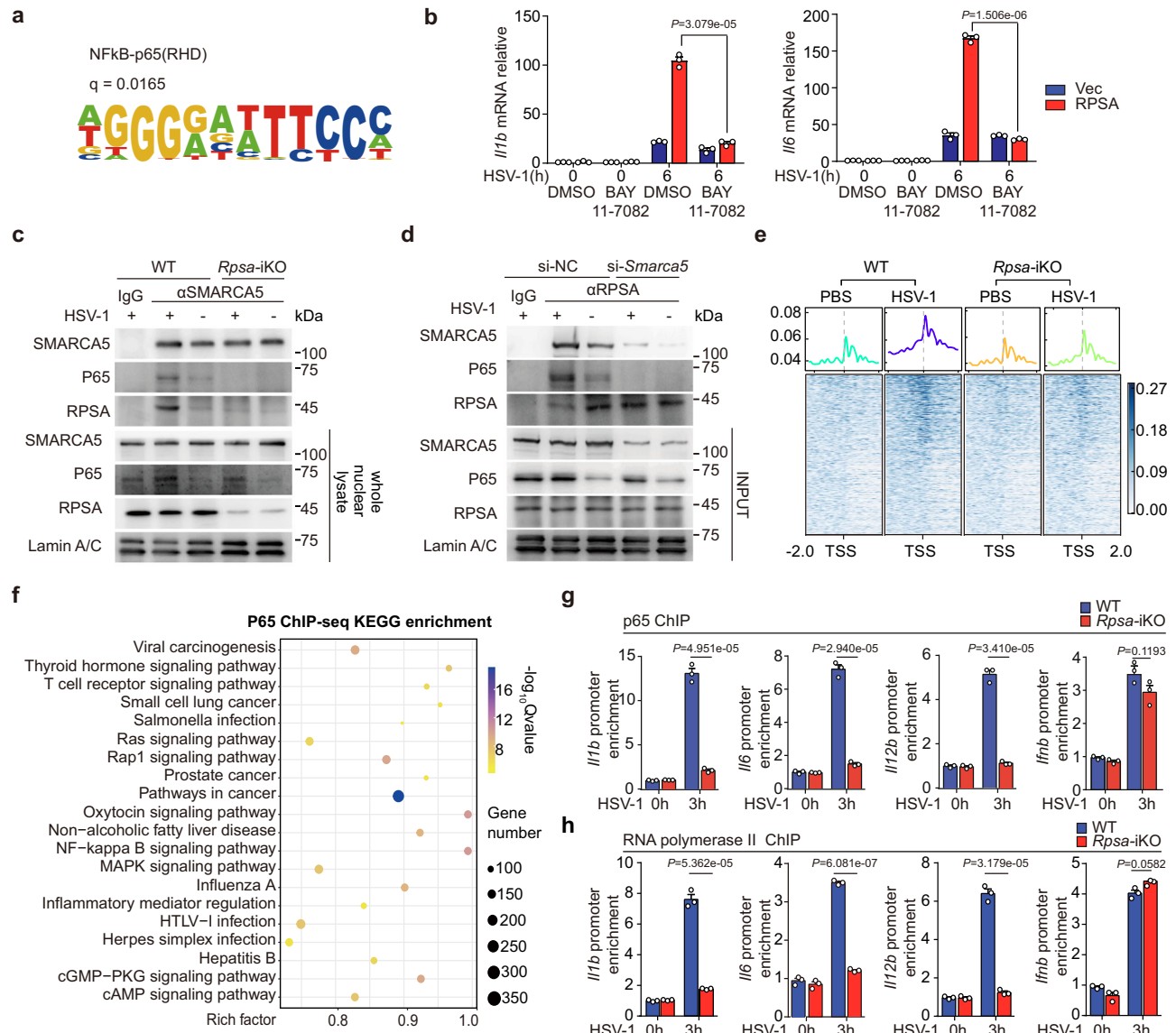

**Fig. 5 | Deficiency of Rpsa reduces P65 subunit binding to the promoter regions of proinflammatory cytokine genes. a** Motif analysis of differential peak enrichment in H3K4me3 ChIP-seq assay of wild-type and *Rpsa*-iKO RAW264.7 cells infected with HSV-1. **b** qRT-PCR analysis for *Il1b* and *Il6* mRNA levels in *Rpsa*-iKO MLE-12 cells transfected with the indicated plasmids. Then, it was pre-treated with DMSO or BAY 11-7082 (10 μM) for 2 h and infected with HSV-1 for 6 h (*n* = 3). **c** Nuclear extracts from wild-type and *Rpsa*-iKO RAW264.7 cells infected with HSV-1 were immunoprecipitated with anti-SMARCA5 antibody or IgG and then immunoblotted for indicated proteins. **d** Mouse PMs were transfected with control siRNA or SMARCA5-specific siRNA for 48 h and then infected with HSV-1. Nuclear extracts were immunoprecipitated with anti-RPSA antibody or IgG. The components in the complex were examined by immunoblot. **e** Distribution of

reads relative to TSS from P65 ChIP-seq assay of wild-type and *Rpsa*-iKO RAW264.7 cells infected with HSV-1. **f** KEGG pathway analysis of genes associated with differential peaks from P65 ChIP-seq assay of wild-type and *Rpsa*-iKO RAW264.7 cells infected with HSV-1. **g** ChIP-qPCR assay of P65 recruitment to *Il1b*, *Il6*, *Il12b* and *Ifnb* promoter regions in wild-type and *Rpsa*-iKO RAW264.7 cells infected with or without HSV-1 (*n* = 3). **h** ChIP-qPCR assay of RNA polymerase II recruitment to *Il1b*, *Il6*, *Il12b*, and *Ifnb* promoter regions in wild-type and *Rpsa*-iKO RAW264.7 cells (*n* = 3). Similar results were obtained for three independent experiments (**b**–**d**, **g**, **h**), and one representative experiment is shown. Data in **b**, **g**, **h** are shown as mean ± s.e.m.. The *P* values were calculated by a two-tailed unpaired Student's *t*-test. Source data are provided as a Source Data file.

Inflammation is integral to the host's response to viral infection. Proinflammatory cytokines are essential in the antiviral host defense[25] by inducing the expression of part of interferon-stimulated genes (ISGs), and retraining viral infection in an interferon-independent manner[26,27]. The timely start of inflammatory response is also vital for facilitating immunocyte recruiting to infection sites. For example, TNF-α and IL-1 are actuators for accelerating adhesion molecule expression on the cell surface, thus enrolling types of immune cells to infection sites to restrain IAV spread[28]. Aside from enhancing pathogen clearance, the continuous inflammatory response also results in immunopathological changes and tissue damage[29,30]. Chronic virus infection can

induce cancer by activating the inflammatory response to stimulate the growth of infected cells and inhibit apoptosis. An aberrant activated inflammatory response correlated with disease severity in patients with SARS-COVID-19 infection[31,32]. In this study, we demonstrated that RPSA resists viral infection by selectively promoting proinflammatory cytokines expression, where we put forward a hypothesis that RPSA may synergize with other innate sensors to fight the virus. Our study also provides potential intervention approaches and therapeutic targets for inflammation-related diseases caused by virus infection.

Several innate sensors have been reported within the cell nucleus. HnRNPA2B1 and IFI16 are nuclear inducers of IFN-I production by

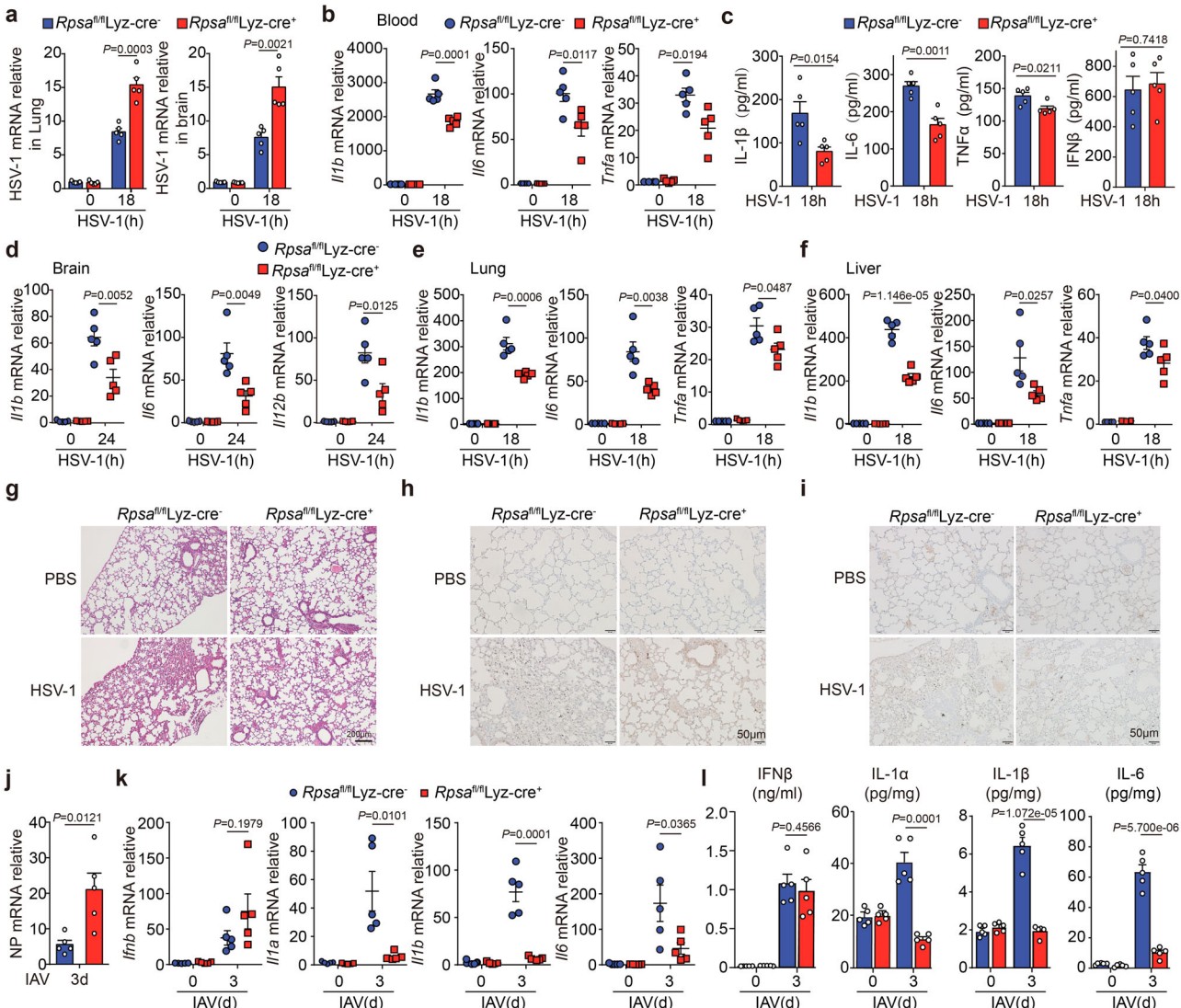

**Fig. 6 | In vivo deficiency of *Rpsa* reduces innate resistance against viral infection. a–i** *Rpsa*fl/fl Lyz-Cre+ mice and the littermates was intravenously (i.v.) challenged with HSV-1 at 8 × 10⁷ plaque-forming units (PFU) (*n* = 5 mouse/group) for the indicated times. qRT-PCR analysis of HSV-1 mRNA in the lung and the brain (**a**). qRT-PCR analysis of *Il1b*, *Il6*, and *Tnfa* mRNAs in peripheral blood (**b**). ELISA assay of IL-1β, IL-6, TNF-α, and IFNβ in peripheral blood (**c**). qRT-PCR analysis of *Il1b*, *Il6*, and *Il12b* mRNAs in the brain (**d**). qRT-PCR analysis of *Il1b*, *Il6*, and *Tnfa* mRNAs in the lung (**e**) and the liver (**f**). HE staining of the lung was shown in (**g**). Immunohistochemical analysis for IL-1β (**h**) and IL-6 (**i**) in the lung. **j–l** *Rpsa*fl/fl Lyz-Cre+ mice and the littermates were intranasally infected with IAV at 100 plaque-forming units (PFU) (*n* = 5 mouse/group). qRT-PCR analysis of IAV NP mRNA in the lung (**j**). qRT-PCR analysis of *Ifnb*, *Il1a*, *Il1b*, and *Il6* mRNAs in the lung (**k**). Protein levels of cytokines in lung homogenate were determined by ELISA (**l**). Similar results were obtained for two independent experiments and one representative experiment is shown. Data in **a–f**, **j–l** shown are shown as mean ± s.e.m.. The *P* values were calculated by a two-tailed unpaired Student's *t*-test. Source data are provided as a Source Data file.

directly recognizing viral DNA in the nucleus[6,11]. Upon sensing viral DNA in the nucleus, hnRNPA2B1 and IFI16 translocate into the cytoplasm and initiate the STING-dependent activation of the TBK1/IRF3 signaling pathway[33]. Nuclear-resident RIG-I recognizes viral RNA in the nucleus, cooperating with its cytoplasmic counterpart to initiate an MAVS-dependent signaling cascade and IFN induction[8]. In this study, we identify that nuclear innate sensor RPSA employed an intra-nuclear way for accurately enhancing host inflammatory response without affecting the cytoplasmic innate immune signaling, which suggests the host has evolved diverse nuclear nucleic acid sensing mechanisms to surveillance invaded pathogens. The cGAS[34] and RIG-I[35] were well reported to recognize the self-genome DNA and self-RNA and lead to pathological states of the host. Whether RPSA is essential for the cell-autonomous inflammatory program is an interesting question. NF-κB is a crucial transcription factor that

fundamentally controls *Ifnb* and proinflammatory cytokine gene expression. During infection, several virus-derived molecular constituents activate the host NF-κB signaling pathway, including glycoproteins recognized by TLR2 on the cell membrane, viral DNA recognized by TLR9-dependent or non-TLR DNA sensors, and dsRNA and ssRNA recognized by TLR3, TLR7/8[36,37] and the non-TLR RNA sensors. Our data showed that RPSA accelerates proinflammatory cytokine gene expression depending on the classical innate immune signaling, those triggered by the cGAS-STING pathway or TLR-TAK1 axis in HSV-1 infections. However, whether RPSA differentially contributes to each classical pathway is potentially to be answered. Besides signaling transduction, recruitment of NF-κB to target loci to subsequently induce transcriptional events is also actively controlled. Our results demonstrate that RPSA directly promotes the activated P65 subunit binding to the proinflammatory cytokine

gene promoters without infecting NF-κB signaling transduction. It would be of interesting to further investigate the crosstalk between nuclear RPSA-controlled and cytoplasm-started inflammatory signals and the involved mechanisms. NF-κB together with IRFs and cJun, are integrated into an intact multiprotein complex known as "enhanceosome", which is assembled on the IFN enhancer and required for initiating transcription[38]. In this study, RPSA was identified to selectively interact with P65 but not IRF3, indicating the regulation of *Ifnb* transcription was independent of this innate nuclear sensor. Moreover, the NF-κB pathway is activated in multiple solid cancers, and crosstalk with a set of signaling pathways, such as P53 and AP1, subsequently enhances cell proliferation and anti-apoptotic gene expression[39]. Whether RPSA is potent for disturbing NF-κB transcriptional activity in cancers still needs to be investigated. The underlying mechanism would raise a therapeutic opportunity.

Posttranslational modifications (PTM) of proteins, particularly phosphorylation, acetylation, and ubiquitination, exert diverse effects on pattern recognition receptor (PRR) -dependent inflammatory responses[40]. Once sensing viral DNA, hnRNPA2B1 demethylated at Arg226 and subsequently initiated IFN-β expression[11]. IFI16 acetylation is required for its innate responses of inflammasome activation and IFN-β production[41]. Deacetylation of RIG-I mediated by HDAC6 is critical for its viral RNA-sensing activity[42]. We find that HSV-1 infection-induced phosphorylation of RPSA on tyrosine 204 is vital for boosting proinflammatory cytokines expression, but the phosphokinase responsible for this phosphorylation remains to be identified. Importantly, constituted activation of RPSA on Y204 promoted *Il-1b* expression and enhanced binding of SMARCA5, even though there were no viral nucleic acids in the presence in the cell compartment (Fig. 4i, j). It suggests this key PTM is critical for RPSA triggering epigenetic modification of target gene sites but may not be responsible for sensing viral nucleic acids. It is urgent to elucidate the underlying mechanism of how RPSA senses nucleic acids.

Epigenetic regulation is a critical way of controlling inflammatory cytokines expression, and has become an intervention strategy for inflammatory-related diseases[43,44]. The ISWI complex ATP-dependently mobilizes nucleosomes and remodels chromatin, thereby regulating the transcription of target genes[45]. The ISWI complex has been shown to enhance NF-κB transcription activity, thus playing a vital role in tumor growth and in the expression of proinflammatory cytokines following virus infection[46]. The nuclear matrix protein scaffold attachment factor A (SAFA) interacts with the ISWI chromatin remodeling complex upon recognition of viral RNA, subsequently improving chromatin accessibility in enhancer and super-enhancer regions of interferon and proinflammatory cytokine genes[47]. In this study, we find that RPSA is required for SMARCA5 complex anchoring on the proinflammatory gene promotors and epigenetically enhancing gene expression, highlighting a crosstalk mechanism between chromatin modifiers and innate immune response. The structure basis of the interaction among RPSA, NF-κB and SMARCA5 is worthy of further investigation. In addition, whether RPSA affects chromatin accessibility in other DNA cis-regulatory elements, such as enhancers or super-enhancers, remains to be further elucidated. Accumulating evidence suggests that epigenetic reprogramming organized "innate immune memory", accompanied by groups of chromatin marks, such as H3K4me3, after HSV-1 infection[48]. There is great interest in RPSA-triggered epigenetic remolding in establishing trained immunity that protects the host against secondary infection.

Ribosomal proteins play an important role in viral infection by participating in the replication and transcription of viral genes and the translation of viral proteins[49]. In addition, ribosomal proteins have been shown to be involved in the regulation of antiviral innate immune signaling pathways[50,51]. RPSA, also known as the 37/67-kDa laminin receptor, has been found to be related to a variety of diseases,

including infections, tumors and neurodegenerative diseases[52,53]. RPSA mutations also lead to congenital asplenia, illustrating the role of RPSA in tissue differentiation[51]. Nuclear and nucleolar localization of RPSA was noted early. It was reported that RPSA serves to sequester the DNA damage repair proteins RNF8 (ring finger protein 8) and BRCA1 (breast cancer 1) to a waiting reserve in the nucleolus[54]. However, the role and underlying mechanisms of RPSA in virus infection and inflammation remain unclear. In this study, we find that the nuclear RPSA interacts with the NF-κB P65 subunit after virus infection and participates in promoting the transcription of proinflammatory cytokine genes. Recently, the chemical compound PAC5 was identified as the agonist of nuclear DNA sensor hnRNPA2B1 to launch the anti-virus effects of the hosts[55]. It is also interesting to explore the protein druggability of the nuclear RPSA, especially to find the druggable pocket around Y204 amino acids. Our study provides valuable insight into the new mechanism in the regulation of nuclear-controlled innate immunity and inflammation by ribosomal protein and deepens our understanding of the non-canonical function of ribosomal proteins in the antiviral innate immune response.

Taken together, we have revealed the function and the mechanism of RPSA in the host response to viral infection. Upon recognizing viral nuclear acid, phosphorylated RPSA regulates the chromatin accessibility by interacting with the ISWI chromatin remodeling complex and activates the expression of inflammatory factors synergistically with activated transcription factor NF-κB. Although there are remaining unsolved questions, RPSA evidently has an important contribution to innate immune defense.

## Methods

### Ethics statement

All animal protocols were approved by the Animal Care and Use Committees of the Institute of Laboratory Animal Science of the Chinese Academy of Medical Sciences (ILAS-GC-2015-002). All mice were bred in specific pathogen-free conditions, housed in cages with five mice per cage, and kept on in a regular 12 h light/12 h dark cycle (lights on at 7:00 am). The temperature was $24 \pm 2\,°C$ and humidity was 40–70%.

### Mice, cells, and reagents

C57BL/6 mice were obtained from Beijing Vital River Laboratory Animal Technology Co., Ltd. (Beijing, China). *Rpsa*^fl/fl mice were generated by the CRISPR–cas9 approach. To establish *Rpsa*-conditional-knockout mice, *Rpsa*^fl/fl mice were crossed with Lyz2-Cre mice. Exons 3 of *Rpsa* were excised by CRE recombinase in myeloid cells. The genotype primers by q-PCR analysis were listed in SI, Table S3.

RAW264.7 cells, A549, MEF cells, MLE-12 cells, HEK293T cells, and Vero cells were obtained from the American Type Culture Collection (ATCC). Mouse BMDMs were prepared by culturing in DMEM medium with 10% FBS and 50 ng/ml of recombinant mouse macrophage colony-stimulating factors (M-CSF; Perprotech). Mouse primary PMs were obtained 3 d after intraperitoneal injection with thioglycolate and cultured with DMEM with 10% FBS.

The *Rpsa*-deficient cells were generated using CRISPR–iCas9 with guide RNA containing plasmids (Table S2). Cas9 is induced by 100 ng/ml of tetracycline (Selleck) for 14 h, the knockout efficiency was determined by immunoblotting. RPSA-overexpressing RAW264.7 cells were generated by transfection with Flag-RPSA-pHAGE vector and then puromycin (Sigma) selection. The reagents and antibodies used in this study are listed in Table S4.

### RNAi, plasmids, and virus

The 20 nM siRNA was transfected into the indicated cells using standard procedures with Lipofectamine RNAiMAX Transfection Reagent (ThermoFisher) according to the manufacturer's instructions. The sequences used for transient silence are shown in Table S1.

The coding sequences of RPSA (gene ID: 16785) and SMARCA5 (gene ID: 93762) with distinct tags were amplified from macrophage complementary DNAs. Then, the CDS of RPSA and SMARCA5 was cloned into a pHAGE vector. Plasmids were transiently transfected into HEK293T cells, MLE-12 cells, or MEF cells with Lipofectamine 3000 Reagent (ThermoFisher) according to the manufacturer's instructions.

HSV-1 was propagated and titrated by the plaque-forming assay on Vero cells. For in vitro challenging, cells were infected with HSV-1 at an MOI of 10. IAV was a gift from Dr. Shuo Liu (Peking Union Medical College, Beijing, China). HSV-1 genome was labeled by adding EdU to the Vero cell medium at 8, 24, and 48 h post-infection. On day 4, the culture supernatant was collected and the labeled virus was purified.

### Cytoplasm and nuclear separation
All nuclear extracts from the cells for immunoprecipitation were prepared using the Nuclear Complex Co-IP Kit (Active Motif) according to the manufacturer's instructions.

### HSV-1 DNA purification, biotin labeling, and nuclear acid affinity purification
The method has been described earlier[11]. Briefly, HSV-1 genomic DNA was purified by using ChargeSwitchg DNA Preparation Kit (Invitrogen) and biotinylated with a biotin 3′-end DNA labeling kit (Pierce Biotechnology). Nuclear extracts were incubated with biotinylated HSV-1 DNA at 4 °C overnight. Then complexes were precipitated on streptavidin-coupled dynabeads and resolved on 10% SDS-PAGE gel.

### EdU labeled HSV-1 viral DNA pull-down assay
The EdU-labeled viral genome DNA pull-down method was described previously[56]. Briefly, mouse BMDMs were infected with unlabeled or EdU-labeled HSV-1 (MOI, 10) for 2 h. To cross-link EdU labeled HSV-1 DNA with its interacting molecules, cells were treated with 1% formaldehyde for 10 min at 4 °C. Unreacted formaldehyde was eliminated with 0.125 M glycine at 4 °C for 10 min. Then, cells were harvested and permeabilized with 0.1% Triton X-100 for 10 min, then washed with PBS. Biotin was connected to the EdU genome via a Click reaction using sequential addition of (+)-sodium-ʟ-ascorbate (10 mM), biotin-TEZ azide (0.1 mM), and copper (II) sulfate (2 mM) for 30 min in dark, then added 1% BSA and 0.5% Tween-20 for 10 min. To separate unsoluble DNA/protein complex, the cells were firstly resuspended in 500 μl of cell lysis buffer (50 mM HEPES, pH 7.8, 0.25% Triton X-100, 0.5% NP-40, 150 mM NaCl, 10% glycerol plus protease inhibitors) and centrifuged at 300 g. Subsequently, the pellet was resuspended in 500 μl RIPA buffer (Thermofisher). DNA was then sheared by sonication and clarified by centrifugation (15,000×g) for 10 min at 4 °C. In total, 1 mg of the extracts were pulled down with 50 μl of streptavidin magnetic beads. Beads with bound complexes reversed protein-DNA cross-linking with elution buffer (Cell Signaling Technology, 14231 s), and proteins were eluted in 1× Laemmlisample buffer (95 °C for 10 min) for immunoblotting.

### DNA and RNA competition assay
HSV-1 genome DNA was biotinylated using a 3′-end DNA labeling kit (Thermofisher) as previously described (11). Recombinant mouse RPSA protein was incubated with biotinylated HSV-1 DNA (5 pmol) without or with concentration-gradient unlabeled HSV-1 DNA (5 pmol, 2 pmol, 1 pmol), naked mouse genome DNA or nucleosomes (Sigma-Aldrich) at 4 °C overnight. The complexes were precipitated on streptavidin-coupled dynabeads and resolved on 10% SDS-PAGE gel.

Recombinant human RPSA protein was incubated with biotinylated HSV-60 (Invivogen) dsDNA at 4 °C overnight. The IAV gRNA was purified using Trizol reagent and biotinylated with a biotin 3′-end RNA labeling kit according to the manufacturer's instructions (Pierce Biotechnology). The complexes were precipitated on streptavidin-coupled dynabeads and resolved on 10% SDS-PAGE gel. For the competition, we used 60 bp HSV-60, 70 bp VACV-70 (Invivogen), poly dA:dT (Invivogen), poly dI:dC (ThermoFisher), G3-YSD (Invivogen), *Escherichia coli* ssDNA (Invivogen) and ISD (Invivogen) for DNA competition. Using unlabeled IAV gRNA for RNA competition.

### Electrophoresis mobility shift assay
In total, 60 bp HSV-60 was biotinylated with a biotin 3′-end DNA labeling kit as a DNA probe. The DNA-binding reaction was carried out with the LightShift Chemiluminescent EMSA kit (Thermo Scientific) according to the manufacturer's instructions. A recombinant mouse RPSA-DNA complex was identified by electrophoresis on a 4% polyacrylamide gel. For the competition, we used 60 bp HSV-60, 70 bp VACV-70 (Invivogen), poly dA:dT, and poly dI:dC (Thermo) unlabeled probes.

### Immunofluorescence microscopy
Mouse BMDMs were seeded on glass chamber slides (ThermoFisher Scientific) infected with or without HSV-1 (MOI, 10). Cells were then washed and blocked using Image-iT signal enhancer (Life Technologies) for 20 min, followed by incubation with primary antibody (Invitrogen, PA5-86634), and then incubated with secondary antibodies conjugated with fluorescent dye. To detect EdU labeled viral genome, cells were fixed, permeabilized, and blocked with Image-iT signal enhancer for 20 min. A CLICK reaction was performed for 30 min at RT using Click-iT EdU reaction additive (Life Technologies), copper sulfate, EdU reaction buffer, and Alexa Fluor 555 azide (ThermoFisher Scientific). To detect EU-labeled IAV, a CLICK reaction was performed using Click-iT EU reaction additive (Life Technologies), copper sulfate, EU reaction buffer, and Alexa Fluor 488 azide according to the manufacturer's instructions (ThermoFisher Scientific). Cells were observed by Olympus FV100MPE microscope, and analyzed with FV10-ASW_Viewer imaging software.

### Nucleofection
Raw264.7 cells were transfected with 1 μg of poly dA:dT (Invivogen) or 1 μg of poly I:C (Invivogen), via Amaxa Nucleofector following the manufacturer's instructions.

### ELISA
Cell culture supernatants from uninfected or virus-infected cells were collected and levels of IL-1α, IFN-β, IL-1β, IL-6, IL-12b, and TNFα secretion were measured with precoated kit (DAKAWE) or microplate (R&D SYSTEMS) follow the manufacturer's protocol.

### Immunoprecipitation (IP)
For immunoprecipitation, the harvested cells were lysed using IP lysis buffer (25 mM Tris-HCl, pH7.5, 150 mM NaCl, 1% NP40, 1 mM EDTA, and protease inhibitor mixture) and 150–200 μg of precleared whole cell lysates or extracted nuclear fractions were incubated overnight with primary antibodies at 4 °C. The antibody from Santa Cruz Biotechnology was used to capture endogenous RPSA. The immune complexes were captured using Protein A/G Magnetic Beads (Thermo Scientific, 26162), washed thrice in 150 mM low salt wash buffer and thrice in 500 mM high salt wash buffer, then examined by immunoblotting.

To detect viral nucleic acids bound to RPSA, the captured immune complexes were treated with a solution of phenol-chloroform-isoamyl alcohol (for DNA, pH = 8.8; for RNA, pH = 4.5). The extracted HSV-1 DNA was detected by PCR. The extracted IAV RNA was detected by PCR after reverse transcription into cDNA.

For unlabeled quantitative proteomics, the captured immune complexes from extracted nuclear fractions were run 1.5 cm into the polyacrylamide gel by electrophoresis and cut off for mass spectrometry detection. For differential binding bands analysis, the captured immune complexes were run into the polyacrylamide gel by

electrophoresis, and the differential binding bands were cut off for mass spectrometry detection after sensitive silver staining.

### DNase I sensitivity assay

Cells were infected with HSV-1 or treated with the corresponding inhibitors. The cells were lysed with 0.1% NP40 in PBS, then treated with DNase I at 37 °C for 30 min and stopped by EDTA. Genome DNA was extracted and subjected to qRT-PCR assay for detection of proinflammatory cytokines promoter regions. The results have been calculated the results with the 2-dCT method relative to those uninfected samples.

### Chromatin immunoprecipitation (ChIP)

ChIP assay was performed using the SimpleChIP Enzymatic Chromatin IP kit (Cell Signaling Technology) according to the manufacturer's instructions. After being infected with HSV-1, cells were cross-linked with 1% formaldehyde and then subjected to nuclear extraction and chromatin digestion with micrococcal nuclease. For immunoprecipitation, digest chromatin was incubated with 7 μg of antibodies overnight at 4 °C with rotation. After that, magnetic beads were added to the immunoprecipitation reaction for 2 h of incubation. After being washed four times with wash buffer, immunoprecipitated chromatin DNA was eluted and quantified by PCR or sequencing (Novogene). Primers used for ChIP quantification are shown in SI, Table 3.

### ATAC-seq

For ATAC-seq, we used the TruePrep DNA Library Prep Kit V2 for Illumina and TruePrep Index Kit V2 for Illumina (Vazyme) according to the manufacturer's instructions. Briefly, the *Rpsa*-iKO and wild-type RAW264.7 cells were seeded and infected with HSV-1(MOI = 10) for 4 h. Then, $5 \times 10^4$ cells were centrifuged at 300 g for 5 mins at 4 °C, then resuspended in 50 μl pre-cold lysis buffer (10 mM Tris-HCl, pH7.4, 10 mM NaCl, 3 mM MgCl$_2$, 0.1% lgepal CA-630) for 10 min on ice. For the transposition reaction, nuclei were collected by centrifugation at 500*g* at 4 °C and resuspended in 50 μl TTE Mix V50 buffer. After reaction at 37 °C for 30 min, the fragments were purified with magnetic beads (Vazyme, N411-01). The transposed DNA fragments then were amplified to build a library after purification for sequencing (Novogene).

### Flow cytometry analysis

Single-cell suspension of splenocytes obtained from 6-week-old *Rpsa*fl/fl Lyz-Cre[+] mice and littermates were labeled with fluorescently labeled antibodies. Then cells were washed with PBS, and immune-cell propagation was analyzed by LSRFortessa (BD Biosciences).

### In vivo viral infection

*Rpsa*fl/fl Lyz-Cre[+] mice and littermates were infected intravenously (i.v.) with $8 \times 10^7$ PFU of HSV-1 viruses. Serum IFN-β, IL-1β, IL-6, TNFα concentrations were determined by ELISA. Total RNA of blood, liver, lung, and brain were extracted with TRIZOL and subjected to a Quantitative RT-PCR assay for detection of proinflammatory cytokine expression.

Mice were infected with IAV (100 PFU) by intranasal delivery for 72 h. The total RNA of the lung was extracted with TRIZOL and subjected to qRT-PCR assay for detection of proinflammatory cytokine expression. IL-1α, IL-1β, IL-6 and IFN-β in lung homogenate supernatant were determined by ELISA.

### Statistical analysis

Results are provided as means ± the standard error. All data are from at least three (in vitro) or two (in vivo) independent experiments. Comparisons between the two groups were performed using a two-tailed unpaired Student's *t*-test. All statistical tests were two-sided, and significance was assigned at $P < 0.05$.

### Reporting summary

Further information on research design is available in the Nature Portfolio Reporting Summary linked to this article.

## Data availability

The RNA seq/ATACseq/ChIP seq data from the present study are deposited in the National Center for Biotechnology Information's Gene Expression Omnibus (GEO) under accession code GSE204895. All other study data are included in the article and/or Supplementary Information. Source data are provided in this paper.

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

## Acknowledgements

We thank Dr. Mingyue Wen and Dr. Lei Wang for the helpful discussion. This work was supported by Grants from the National Natural Science Foundation of China (82388201 to X.C., 32100728 to H.X.) and the Chinese Academy of Medical Sciences Innovation Fund for Medical Sciences (2021-I2M-1-017 to X.C.).

## Author contributions

X.C. designed the experimental approach and supervised the study; Y.J., S.S., Y.Q., X.W., Y.Y., X.Z., Y.Z., Y.L., B.W. and H.X. performed the detection of RT-PCR, Western blot analysis, ELISA, and animal experiments; Y.J. and H.X. performed flow cytometry analysis; Y.J., H.X. and X.C. analyzed data and wrote the paper.

## Competing interests

The authors declare no competing interests.
