## [Peer Review File · Nature Communications]

Reviewers' Comments:

Reviewer #1:

Remarks to the Author:

In the present study, Jiang and colleagues suggest RPSA as a nuclear sensor of viral nucleic acids and propose a mechanism of RPSA-induced virus-infection triggered proinflammatory cytokine activation in cells and in vivo in mice. The authors demonstrated association of RPSA with viral nucleic acids upon infection and showed direct binding of RPSA to HSV-1 DNA by various methods supporting the idea that RPSA is a direct sensor of nuclear replicating viruses. Virus-infection induced phosphorylation at residue Tyr204 promotes RPSA interaction with the chromatin remodeling complex SMARCA5, therefore increasing chromatin accessibility and subsequent proinflammatory cytokine expression. In summary, the authors identified a novel nuclear innate sensor and proposed a unique and interesting short-cut mechanism enhancing the inflammatory response through epigenetic modification by RPSA.

This is a carefully executed study with conclusions well supported by the data. The findings are novel and of high interest to a wider audience. The manuscript provides sound evidence for the identified sensor and propose a unique mechanism for immune defense. To my point of view, there is no additional evidence needed. The following points could be considered to make an excellent study even more convincing.

1) The authors show convincing results on RPSA binding to nucleic acids. Furthermore, infection induced Tyr204 phosphorylation followed by binding to SMARCA5. It would be interesting to know whether binding of viral nucleic acids induce the phosphorylation event? This would connect the two major findings.

2) RPSA promotes proinflammatory cytokine gene transcription upon infection of nuclear replicating viruses, Influenza and HSV. So, both, viral DNA and RNA seem to be recognized by RPSA. The authors should consider proving binding of RPSA to RNA/dsRNA, not only to dsDNA or could discuss their hypothesis on how RPSA could sense both RNA and DNA.

Minor point:

Nucleic acid is misspelled in the title and most part of the manuscript.

Reviewer #2:

Remarks to the Author:

In this manuscript, the author investigate the role of the ribosomal protein RPSA in the activation of inflammatory gene expression after virus infection. The authors provide evidence that RPSA enhances the expression of pro-inflammatory cytokine genes driven by NF- κ B after infection with HSV-1 and influenza A virus, while not affecting the interferon response. It is shown that the expression of pro-inflammatory cytokines is enhanced via RPSA interacting with SMARCA5 and enhancing chromatin accessibility for NF- κ B p65. The authors propose that RSPA acts as a direct sensor of HSV-1 DNA (and possibly influenza virus RNA) as well as an effector which promotes chromatin accessibility. However, as it stands it is unclear how the proposed sensing activity integrates with well-established nucleic acid sensors during viral infection.

Overall, the authors provide convincing evidence that RPSA promotes virus-induced cytokine expression by enhancing chromatin accessibility. While the authors provide evidence that RPSA can bind DNA, it is currently not entirely clear that DNA binding can be equated with sensing per se. However, even if further experiments were to show that RPSA transduces the signal from STING or MAVS for instance to promote NF- κ B-mediated transcription, rather than acting as a sensor in its own right, this would still be an important finding in the field.

Specific points:

- 1.) Evidence should be provided at the start that the absence of RPSA does not affect HSV-1 and IAV-1 entry and replication etc. For instance, viral mRNAs could be quantified alongside the qPCR analysis in Fig. 1 a-c to show that RPSA deletion affects cytokine expression, but not virus levels in the cells. Showing the (unchanged) levels of IFN- β in Fig. 1 already would support this notion.
- 2.) It is shown that RPSA affects the expression of pro-inflammatory cytokines during both HSV-1

and IAV infection. Does RPSA also bind IAV RNA, and co-localise with IAV viral factories, as would be expected if it indeed acts as a sensor?

3.) Currently there is not much information on the DNA (and/or RNA) binding activity in the RPSA protein, which would allow to dissect its role in the sensing pathway. Could structural predictions be used to identify a binding site and predict residues/fragments for mutagenesis analysis?

4.) In Fig. 2a-b, it is shown that loss of RPSA decreases cytokine mRNA and protein levels, but does not abolish the induction completely. It would be interesting to see whether the cytokine response to HSV-1 in these is completely or partially cGAS- and STING-dependent (using deletion cells, RNAi or inhibitors). This would provide some clues as to whether there is redundancy between two sensors, or whether RPSA boost responses that are mainly STING-dependent.

5.) In Fig. 2e it would be nice to see typical anti-viral response genes including type I interferons, CXCL10 and CCL5 (which may not be affected by RPSA deletion) – this would strengthen the argument that infection proceeds as normal, but one arm of the innate immune response is specifically affected.

6.) The experiments with nuclear and cytoplasmic transfection in Fig. 2f-h are misleading. For “nuclear” transfection, only dsDNA is used, while “cytoplasmic” transfection the stimulus is CpG DNA and poly(I:C). The same stimuli should be used with the two different transfection methods to show whether there are any differences in terms of localisation, rather than nucleic acid type.

7.) In this context it would also be interesting to examine whether cytokine production in response to a STING agonist is affected. If it is, this would support a pathway where RPSA is activated as another effector mechanism of STING signalling, rather than a sensor (which would also be interesting).

8.) The fractionation in S4b is not convincing – maybe immunofluorescence could be used to quantify p65 translocation.

9.) The phosphorylation of RPSA is intriguing, and points towards the involvement of additional proteins in the activation/regulation of this response. To integrate RPSA into known signalling cascades phosphorylation could be tested in dependence of STING signalling (for HSV-1) or MAVS (for IAV), using inhibitors, RNAi or deletion cells.

10.) Another way to test whether RPSA has direct sensing function, would be to test SMARCA5 interaction and chromatin accessibility in for instance STING deletion cells. As many proteins in the cell have DNA binding activity and indeed this activity may be needed for influencing chromatin accessibility, it would be important to show whether this indeed contributes to “sensing” independently. Alternatively, it might be possible to integrate RPSA into known signalling cascades (particularly cGAS-STING signalling for DNA viruses).

11.) Additional experiments with STING inhibitor (or deletion) and Takinib separately would be more informative than the current “pan-inhibitor” experiments.

12.) Very little information is provided about an involvement of RPSA in human cells – this would enhance the translatability of the findings.

Minor points:

13.) Please include cytokine levels for mock infections / stimulations where possible, e.g. in Fig. 1a, 3a, 5b, 6l etc.

14.) The lay-out of some of the graphs could be more consistent, e.g. uninfected – infected, DMSO – inhibitor (e.g. in Fig. 3). The labelling in Fig. 6 is very small and unclear.

15.) Fig. 5 which shows an involvement of p65 could be moved to earlier (after Fig. 3), as many of the genes shown are well known to be NF- κ B dependent.

Reviewer #3:

Remarks to the Author:

Jiang et al. report on the role of RPSA during innate immune responses, finding the protein acts as a nuclear sensor of viral nucleic acids that in turn modulates gene regulatory responses as part of the innate immune response. More specifically, they identify that virally-mediated phosphorylation of residue Tyr204 on RPSA causes recruitment of the SMARCA5/ISWI complex to NF κ B proinflammatory promoters independent of p65 activation to increase chromatin accessibility and gene transcription.

This is a compelling paper with a lot of data, and potentially identifies an important, alternative

host defense sensor and pathway that appears independent of traditional pattern recognition receptor pathways but nonetheless modulates their activity by integrating regulatory information at the site of chromatin. While the potential significance of the work is high, and many of the experiments show promising and interesting results, some of the work and its interpretation are stretched thin at times leading to unsubstantiated conclusions. Suggestions to improve the manuscript:

One of the main issues with the manuscript is that it mixes and matches data types, experiments, and interpretation. For example, several NGS profiling experiments are performed in genome-wide fashion (e.g. ATAC-seq, H3K4me3 ChIP-seq, p65 ChIP-seq) while several others are not (ChIP q-PCR for RPSA, SMARCA5, RNAPII). Given the claims of the authors, I would highly recommend performing ChIP-seq with RPSA and SMARCA5 to fully understand the extent to which these proteins co-occupy chromatin and at which targets they localize to – for example, how specific are RPSA/SMARCA5 for NFkB targets? A more integrated view of how NFkB, RPSA, SMARCA5, and chromatin accessibility relate to one another would help solidify the mechanistic conclusions put forth by the authors.

With respect to the text near the call-out to Fig. 3c/d, the authors claim that Rpsa-iKO have less DNA accessibility near TSS regions, but Fig.3c does not seem to convincingly indicate this – at least it is not clear looking at the figure. Also, it is very odd that DNA promoter accessibility is shown to decrease in both WT and Rpsa-iKO samples during HSV infection even though the promoters of the genes shown are dramatically up-regulated (Fig. 3d). These observations are extra confusing because it seems that the Rpsa-iKO samples show MORE DNase accessibility on pro-inflammatory promoters, where the model proposed would suggest there should be less. I would highly recommend expanding the genome-wide profiling to include ATAC-seq at both control and infected conditions to include a more comprehensive analysis to make this data more robust instead of relying on the candidate promoter DNase hypersensitivity data.

A more detailed description of the RNA-seq data is needed. For example, Supp. Fig 3b is hard to process as it contains a mix of up- and down- regulated genes relative to each of the experimental groups mentioned. I might include a global clustering/heatmap of the experiment (like shown in Fig. 2e) that provides a birds-eye-view of the gene expression changes that occur in the experiment to provide additional context. For example, it seems clear that many proinflammatory genes have reduced induction when Rpsa is depleted, but are there also a significant number of genes that are perhaps hyper-induced too?

In Figs. 1 & 2 the authors make the claim that RPSA is a nuclear sensor for viral RNA/DNA. While several insightful experiments were performed (i.e. nuclear transfection of dsDNA), additional controls should also be considered. For example, is RPSA functional during the innate response to non-nuclear viruses (e.g. flaviviruses) or other pattern recognition receptor signals/NFkB signaling pathways (e.g. Tlr4 pathway activation/TNF)? These would appear to be important controls. Fig. 2g-h help address this, but additional stimuli would help make this more convincing. In addition, given the model that RPSA is needed to help increase pro-inflammatory gene expression during these responses, what is compensating its role in driving inflammatory gene expression in the case of cytoplasmic activation by e.g. polyIC where RPSA is presumably not activated?

The following statement is not well supported in the literature: "Chromatin remodeling complexes increase chromatin accessibility to maintain transcriptional activity, which are marked by trimethylation of histone H3 at lysine 4 (H3K4me3) around TSS of target genes". Certainly, H3K4me3 is associated with gene TSS and active transcription, but H3K4me3 is not in and of itself proof of ISWI chromatin remodeling, as many locations where chromatin remodelers are recruited (i.e. enhancers) do not necessarily have this chromatin mark. Does Fig. 3f show all genes in the genome? Would this imply there is a global decrease in H3K4me3? How does that impact the interpretation of the data?

The ChIP-seq profile shown in Fig. 5d is a bit concerning – most ChIP-seq data for transcription factors, including previously published p65 ChIP-seq experiments, show maximal enrichment for the TF just upstream of the TSS in the open chromatin region of the promoter. These experiments

seem to show a distribution of ChIP-seq signal just downstream of the TSS, more similar to that of a ChIP-seq experiment like H3K4me3 or H3K27ac.

The authors mention a multitude of targets from their proteomics data, but do not report the data itself. Not only does this limit the interpretability of their results, but it is also unclear exactly how the interaction with SMARCA5 fits into those results – was it by far one of the strongest interactions, or are there a multitude of other proteins that interact with RPSA with similar strength?

The association of RPSA with SMARCA5 is intriguing and presents the exciting possibility that RPSA is critically involved in DNA accessibility. However, one problem is that DNA accessibility (ATAC-seq), RNAPII and TF recruitment, and chromatin marks like H3K4me3 are often highly correlated with one another and transcription. As a result, it can be hard to definitely attribute specific mechanisms when looking at changes in these data types, particularly when they change in unison as e.g. a loss of ATAC-seq signal could simply mean that the transcription of the gene is down. In addition to thinking about how the nuance that this might introduce into their interpretation of the results, I might recommend the authors consider a more targeted in vitro chromatin remodeling assay to assess RPSA's role in modulating ISWI activity.

Have the authors considered looking at these mechanisms regulate gene expression on the viral genome? Does RPSA/SMARCA5 etc. get recruited the HSV genome? Does the RPSA-iKO show strong modulation of viral gene expression for HSV or IAV?

Another suggestion for a relatively complicated manuscript such as this is to include a schematic/model that helps summarize the findings as a final figure panel or supplementary figure.

Minor comments:

I would consider rewording the sentence in the abstract: “nuclear short way” does not seem very concise – I recommend replacing the term with a better description.

Many mentions of “viral nuclear acids” – maybe the authors mean viral nucleic acids?

Consider including MDA/IFIH1 in the 3rd sentence for more completeness.

Although the writing in the manuscript is relatively easy to understand, there are lots of minor language problems – needs to be carefully edited before publication.

Might be nice to comment on RPSA's previously known roles and alternative functions (and maybe how the protein was originally named).

Point-by-point Response

To Reviewer #1:

In the present study, Jiang and colleagues suggest RPSA as a nuclear sensor of viral nucleic acids and propose a mechanism of RPSA-induced virus-infection triggered proinflammatory cytokine activation in cells and in vivo in mice. The authors demonstrated association of RPSA with viral nucleic acids upon infection and showed direct binding of RPSA to HSV-1 DNA by various methods supporting the idea that RPSA is a direct sensor of nuclear replicating viruses. Virus-infection induced phosphorylation at residue Tyr204 promotes RPSA interaction with the chromatin remodeling complex SMARCA5, therefore increasing chromatin accessibility and subsequent proinflammatory cytokine expression. In summary, the authors identified a novel nuclear innate sensor and proposed a unique and interesting short-cut mechanism enhancing the inflammatory response through epigenetic modification by RPSA.

This is a carefully executed study with conclusions well supported by the data. The findings are novel and of high interest to a wider audience. The manuscript provides sound evidence for the identified sensor and propose a unique mechanism for immune defense. To my point of view, there is no additional evidence needed. The following points could be considered to make an excellent study even more convincing.

Response: Thanks for the positive comments. Following the comments and insightful suggestions, we provided additional data to address the concerns and further improve the quality of this work.

1) The authors show convincing results on RPSA binding to nucleic acids. Furthermore, infection induced Tyr204 phosphorylation followed by binding to SMARCA5. It would be interesting to know whether binding of viral nucleic acids induce the phosphorylation event? This would connect the two major findings.

Response: Thanks for the insightful suggestions. Our previous results in **Fig. 4f** and **g**

showed that the nucleus-located RPSA was only phosphorylated upon HSV-1 or IAV infection, suggesting the presence of viral nucleic acids is especially important for the phosphorylation event of RPSA. To further address this critical concern, we established the cytoplasm replicated virus VACV (Vaccinia virus) infection model, and our results showed that VACV infection did not enhance the phosphorylation of nuclear RPSA (**Revised Supplementary Fig. 6g**). Moreover, we also stimulated RAW264.7 cells with cGAMP to initiate innate signals from the cytoplasm then tested the phosphorylation level of nuclear RPSA, and found cGAMP stimulation could not induce such a PTM as HSV-1 or IAV virus infection induced (**Revised Supplementary Fig. 6e**). Taken together, our results show that the nuclear RPSA binding of viral nucleic acids mediates RPSA phosphorylation and subsequently boosts transcription events.

2) RPSA promotes proinflammatory cytokine gene transcription upon infection of nuclear replicating viruses, Influenza and HSV. So, both, viral DNA and RNA seem to be recognized by RPSA. The authors should consider proving binding of RPSA to RNA/dsRNA, not only to dsDNA or could discuss their hypothesis on how RPSA could sense both RNA and DNA.

Response: Our previous RNA immunoprecipitation analysis in the nuclear fraction of IAV-infected BMDM indicated the interaction between IAV genome RNA (gRNA) and nuclear RPSA (**Fig.1e**). To further clarify the direct interaction between IAV genomic RNA and RPSA, we labeled IAV gRNA with biotin, then the RNA pulldown assay was carried out in **Revised Fig. 11**. Our results showed the direct interaction between RPSA and viral gRNA. Strikingly, the immunofluorescence assay exhibited that RPSA co-localized with EU-labeled IAV gRNA within A549 cell nucleus (**Revised Fig.1g**). So, we conclude in the Results of our revised manuscript that nuclear RPSA is a viral nucleic acids-binding molecule.

Minor point: *Nucleic acid is misspelled in the title and most part of the manuscript.*

Response: We corrected it.

To Reviewer #2:

In this manuscript, the author investigate the role of the ribosomal protein RPSA in the activation of inflammatory gene expression after virus infection. The authors provide evidence that RPSA enhances the expression of pro-inflammatory cytokine genes driven by NF- κ B after infection with HSV-1 and influenza A virus, while not affecting the interferon response. It is shown that the expression of pro-inflammatory cytokines is enhanced via RPSA interacting with SMARCA5 and enhancing chromatin accessibility for NF- κ B p65. The authors propose that RSPA acts as a direct sensor of HSV-1 DNA (and possibly influenza virus RNA) as well as an effector which promotes chromatin accessibility. However, as it stands it is unclear how the proposed sensing activity integrates with well-established nucleic acid sensors during viral infection.

Overall, the authors provide convincing evidence that RPSA promotes virus-induced cytokine expression by enhancing chromatin accessibility. While the authors provide evidence that RPSA can bind DNA, it is currently not entirely clear that DNA binding can be equated with sensing per se. However, even if further experiments were to show that RPSA transduces the signal from STING or MAVS for instance to promote NF- κ B-mediated transcription, rather than acting as a sensor in its own right, this would still be an important finding in the field.

Response: Many thanks for the insightful suggestions, which are very helpful to us for further improving the quality of this work. To clarify the uncertainties, we performed additional experiments to address the concerns and tried efforts to make this work more compelling. We provided a set of experimental evidence to elucidate that RPSA senses nucleus-located pathogenic DNA and RNA and then boosts proinflammatory cytokine gene expression at epigenetic levels (**Revised Fig. 1b, 1c, 1g, 1i; Fig. 2g, 2h; Fig. 3a; Fig. 4c; Supplementary Fig. 1c, Supplementary Fig. 3g, Supplementary Fig. 5a, 5c, 5d, Supplementary Fig. 6e, 6g and Supplementary Fig. 7b**). RPSA enhances proinflammatory cytokine gene expression depending on NF- κ B activation, thus compensating the established cytoplasm innate signal cascade.

Specific points:

1.) Evidence should be provided at the start that the absence of RPSA does not affect HSV-1 and IAV-1 entry and replication etc. For instance, viral mRNAs could be quantified alongside the qPCR analysis in Fig. 1 a-c to show that RPSA deletion affects cytokine expression, but not virus levels in the cells. Showing the (unchanged) levels of IFN- β in Fig. 1 already would support this notion.

Response: We quantified viral gDNA levels in peritoneal macrophages upon HSV-1 infection and demonstrated that the knocking down RPSA did not affect HSV-1 entry (**Revised Supplementary Fig. 1c**). We also quantified both HSV-1 and IAV mRNA levels in the RPSA deleted- and control A549 cells upon infection. These results showed that RPSA deficiency did not affect intracellular viral entry and replications (**Revised Fig. 1b-c**).

2.) It is shown that RPSA affects the expression of pro-inflammatory cytokines during both HSV-1 and IAV infection. Does RPSA also bind IAV RNA, and co-localise with IAV viral factories, as would be expected if it indeed acts as a sensor?

Response: Our previous RNA immunoprecipitation analysis in the nuclear fraction of IAV-infected BMDM indicated the interaction between IAV gRNA and nuclear RPSA (**Fig.1e**). To further clarify the direct interaction between IAV genomic RNA (gRNA) and RPSA, we labeled IAV gRNA with biotin. Then the RNA pulldown assay was carried out, as shown in **Revised Fig. 1l**. Our result showed the direct interaction between RPSA and IAV gRNA. Strikingly, the immunofluorescence assay exhibited that RPSA co-localized with EU-labeled IAV gRNA within A549 cell nucleus (**Revised Fig.1g**).

3.) Currently there is not much information on the DNA (and/or RNA) binding activity in the RSPA protein, which would allow to dissect its role in the sensing pathway. Could structural predictions be used to identify a binding site and predict

residues/fragments for mutagenesis analysis?

Response: Our previous data have shown that both the N-terminal and C-terminal of RPSA were necessary for its binding to HSV-1 gDNA (**Fig. 1m**). We tried to predict the interface between RPSA and HSV-60 dsDNA using HDOCK server (<http://hdock.phys.hust.edu.cn/>) (Yan, Y. *et al.* The HDOCK server for integrated protein-protein docking. *Nat. Protoc.* 2020 May; 15(5):1829-1852.). As shown below, the RPSA has a comparable level of docking and confident score with cGAS, as well as a mid ligand rmsd value between the well know DNA sensors, cGAS and SAFA (**See Figure for reviewer below Fig. a**). In the top1 predicted model, the RPSA inserts into the DNA groove with two critical amino acids arms (**See Figure for reviewer below Fig. b**), located separately with the N- or C-terminal of RPSA. It is interesting to investigate and prove the functions of these two arms in sensing nucleic acids in the future.

Figure for reviewer. Structural prediction of RPSA and HSV-60 complex. **a** docking scores of RPSA, cGAS and SAFA. **b** binding model between RPSA and HSV-60 DNA.

4.) In Fig. 2a-b, it is shown that loss of RPSA decreases cytokine mRNA and protein levels, but does not abolish the induction completely. It would be interesting to see whether the cytokine response to HSV-1 in these is completely or partially cGAS- and STING-dependent (using deletion cells, RNAi or inhibitors). This would provide some clues as to whether there is redundancy between two sensors, or whether RPSA boost responses that are mainly STING-dependent.

Response: Viral entry of host cells can immediately trigger proinflammatory gene

expression by activating NF- κ B and MAPK signaling cascades, mainly depending on TLR-TAK1 and cGAS-STING axis. Consistently, as shown in **Revised Fig. 3a**, both pathways contribute to initiating proinflammatory gene expression. The overexpressed RPSA restored the proinflammatory cytokine gene expression when the inhibitors blocked one of the classically recognized signaling cascades, indicating RPSA depends on and made up for the cytoplasm innate signal. Strikingly, RPSA overexpression no longer rescued the proinflammatory gene expression when we used the combined inhibitors. Moreover, inhibition of STING in peritoneal macrophages partially decreased cytokine mRNA level, and the knocking down of RPSA further reduced the mRNA level (**Revised Supplementary Fig. 5a**). Collectively, our data indicated that RPSA is partially dependent on the innate signal cascade from the cGAS-STING pathway. Importantly, these two sensor pathways play nonredundancy roles in defending against viral infection.

5.) *In Fig. 2e it would be nice to see typical anti-viral response genes including type I interferons, CXCL10 and CCL5 (which may not be affected by RPSA deletion) – this would strengthen the argument that infection proceeds as normal, but one arm of the innate immune response is specifically affected.*

Response: Many chemokine genes have also been transcribed by NF κ B, such as *Cxcl10* (Xia, J. B. *et al.*, Hypoxia/ischemia promotes CXCL10 expression in cardiac microvascular endothelial cells by NF κ B activation. *Cytokine* **81**, 63-70 (2016); Albarnaz J. D. *et al.*, Molecular mimicry of NF-kappaB by vaccinia virus protein enables selective inhibition of antiviral responses. *Nat. Microbiol.* **7**, 154-168 (2022).) and *Cxcl2* (Liu, S. *et al.*, Cis-acting lnc-Cxcl2 restrains neutrophil-mediated lung inflammation by inhibiting epithelial cell CXCL2 expression in virus infection. *Proc. Natl. Acad. Sci. U. S. A.* **118**, (2021)). So, we checked the expression of other ISGs, which were predominantly induced by interferons. As shown in **Revised Supplementary Fig. 1c**, the knocking-down of *Rpsa* in peritoneal macrophages did not affect the expression of *Ifit1* and *Rsad2*.

6.) *The experiments with nuclear and cytoplasmic transfection in Fig. 2f-h are misleading. For “nuclear” transfection, only dsDNA is used, while “cytoplasmic” transfection the stimulus is CpG DNA and poly (I:C). The same stimuli should be used with the two different transfection methods to show whether there are any differences in terms of localisation, rather than nucleic acid type.*

Response: We rechecked the proinflammatory gene expression with nuclear-transfected poly I:C (**Revised Fig. 2g**) or cytoplasm-transfected dsDNA (**Revised Fig. 2h**) in RPSA inducible knockout (iKO) and the control RAW264.7 cells. In consistent with our previous conclusions, the presence of viral nucleic acids in the host nucleus is necessary for activating RPSA.

7.) *In this context it would also be interesting to examine whether cytokine production in response to a STING agonist is affected. If it is, this would support a pathway where RPSA is activated as another effector mechanism of STING signalling, rather than a sensor (which would also be interesting).*

Response: We stimulated RPSA-stable overexpressed RAW264.7 and the wild-type cells with cGAMP and then determined proinflammatory cytokine gene expression. The results showed that the loss of RPSA did not affect cGAMP-induced cytokine mRNA expressions (**Revised Supplementary Fig. 3g**).

8.) *The fractionation in S4b is not convincing – maybe immunofluorescence could be used to quantify p65 translocation.*

Response: We checked P65 translocation upon HSV-1 stimulation using the immunofluorescence method, as shown in **Revised Supplementary Fig. 4c**.

9.) *The phosphorylation of RPSA is intriguing, and points towards the involvement of additional proteins in the activation/regulation of this response. To integrate RPSA into known signalling cascades phosphorylation could be tested in dependence of STING signalling (for HSV-1) or MAVS (for IAV), using inhibitors, RNAi or deletion*

cells.

Response: We checked the tyrosine phosphorylation events of RPSA in both cGAMP-stimulated RAW264.7 cell nuclear lysate (**Revised Supplementary Fig. 6e**) and DNA virus VACV (Vaccinia virus, cytoplasm-replicated)-infected HEK293T cells (**Revised Supplementary Fig. 6g**). The results showed that neither stimuli could induce tyrosine phosphorylation of nuclear RPSA. Together with the previous results in **Fig. 4f** and **4g**, these results demonstrated that the enhanced tyrosine phosphorylation on RPSA was largely dependent on pathogenic nucleic acids present in host cell nucleus.

Given the nucleus-located RPSA was robustly phosphorylated on Y204 in HSV-1 or IAV infection and subsequently mediated interaction with SMARCA5 for boosting gene transcription, we strongly considered some phosphokinases located in host nucleus or even within the chromosome remodeling complex, were responsible for this phosphorylation events. It is interesting to identify the critical factor(s) in the future.

10.) Another way to test whether RPSA has direct sensing function, would be to test SMARCA5 interaction and chromatin accessibility in for instance STING deletion cells. As many proteins in the cell have DNA binding activity and indeed this activity may be needed for influencing chromatin accessibility, it would be important to show whether this indeed contributes to “sensing” independently. Alternatively, it might be possible to integrate RPSA into known signalling cascades (particularly cGAS-STING signalling for DNA viruses).

Response: We checked chromatin accessibility in RPSA iKO RAW264.7 cells which were pretreated with or without STING inhibitor before infection with HSV-1 (**Revised Supplementary Fig. 5c**). The results showed that the inhibitor treatment had little effect on DNase sensitivity. Accompanying the DNase I accessibility assays in cGAMP-stimulated RPSA iKO and the wild-type controls (**Revised Supplementary Fig. 5d**), we proposed that RPSA promoted the chromosome remodeling was STING independency. Strikingly, in the VACV infection model, virus

infection neither enhanced the interactions among RPSA, SMARCA5 and P65 (**Revised Supplementary Fig. 7a**), demonstrating that sensing of pathogenic nucleic acids in the host nucleus is essential for activating RPSA. Together with the direct interaction between RPSA and viral nucleic acids, our results demonstrate that nuclear RPSA acts as a sensor that accelerates proinflammatory cytokine gene expression under an epigenetic mechanism.

11.) Additional experiments with STING inhibitor (or deletion) and Takinib separately would be more informative than the current “pan-inhibitor” experiments.

Response: Many thanks for the suggestion. Initially, we wondered whether RPSA boosted proinflammatory factor mRNA expression depending on the innate signals from the cytoplasm and, if so, which one would be the critical pathway. We checked the contribution of each path by adding the individual inhibitor and the combined inhibitors (pan-inhibitors). As the results shown in **Revised Fig.3a**, RPSA restored the proinflammatory cytokine gene expression when blocking one of the pathways, suggesting RPSA did depend on and make up for the cytoplasm-derived innate signal. However, we need to figure out whether and how RPSA is selective based on current results. It would be interesting to systematically figure out the relationship between RPSA and these signaling pathways, as well as the contribution of each signaling pathway in the future.

12.) Very little information is provided about an involvement of RPSA in human cells – this would enhance the translatability of the findings.

Response: To provide solid evidence of the involvement of RPSA in human infection diseases, we constructed the RPSA-deleted human A549 epithelial cells. By testing the expression of pro-inflammatory cytokine genes in both HSV-1 and IAV infection models, we demonstrated that RPSA also enhanced the pro-inflammatory and some NFkB-regulated chemokine genes expression without disturbing the *Ifnb* gene expression in human cells (**Revised Fig.1b-c**).

Minor points:

13.) Please include cytokine levels for mock infections / stimulations where possible, e.g. in Fig. 1a, 3a, 5b, 6lc et.

Response: We checked and included mock infection data in all figures, including **Revised Fig.1a, 3a, 3c, 3f, 5b and 6l.**

14.) The lay-out of some of the graphs could be more consistent, e.g. uninfected – infected, DMSO – inhibitor (e.g. in Fig. 3). The labelling in Fig. 6 is very small and unclear.

Response: We reconciled the lay-out of all the graphs and enlarged the symbol of each data point in **Fig. 6.**

15.) Fig. 5 which shows an involvement of p65 could be moved to earlier (after Fig. 3), as many of the genes shown are well known to be NF- κ B dependent.

Response: Many thanks for the suggestion. However, in **Fig.5**, we showed the H3K4me3 ChIP-seq motif enrichment and the interaction relationship among RPSA, NF- κ B and SMARCA5. As we identified SMARCA5 by unlabeled quantitative proteomics analysis, it is better to show this identification data earlier.

To Reviewer #3:

Jiang et al. report on the role of RPSA during innate immune responses, finding the protein acts as a nuclear sensor of viral nucleic acids that in turn modulates gene regulatory responses as part of the innate immune response. More specifically, they identify that virally-mediated phosphorylation of residue Tyr204 on RPSA causes recruitment of the SMARCA5/ISWI complex to NF κ B proinflammatory promoters independent of p65 activation to increase chromatin accessibility and gene transcription.

This is a compelling paper with a lot of data, and potentially identifies an important, alternative host defense sensor and pathway that appears independent of traditional pattern recognition receptor pathways but nonetheless modulates their activity by integrating regulatory information at the site of chromatin. While the potential significance of the work is high, and many of the experiments show promising and interesting results, some of the work and its interpretation are stretched thin at times leading to unsubstantiated conclusions. Suggestions to improve the manuscript:

1) One of the main issues with the manuscript is that it mixes and matches data types, experiments, and interpretation. For example, several NGS profiling experiments are performed in genome-wide fashion (e.g. ATAC-seq, H3K4me3 ChIP-seq, p65 ChIP-seq) while several others are not (ChIP q-PCR for RPSA, SMARCA5, RNAPII). Given the claims of the authors, I would highly recommend performing ChIP-seq with RPSA and SMARCA5 to fully understand the extent to which these proteins co-occupy chromatin and at which targets they localize to – for example, how specific are RPSA/SMARCA5 for NFkB targets? A more integrated view of how NFkB, RPSA, SMARCA5, and chromatin accessibility relate to one another would help solidify the mechanistic conclusions put forth by the authors.

Response: We tried to perform the ChIP-seq experiment in flag-RPSA over-expressing RAW264.7 cells infected with or without HSV-1 using the anti-DYKDDDDK antibody (Cat. No. 14793, CST) in two independent experiments. However, we found the NGS data displayed a high background in this cell line using this anti-DYKDDDDK antibody. It is barely to analyze the data in high quality; even a weak signal of differential peaks could be enriched in the NFkB and inflammation-related pathways (**Supplementary Fig.4e**). We then considered the conditions in two aspects: (1) RPSA was not so “directly” binding to the packages genome. It only interacted with the histones or chromosome remodeling complex. During the ChIP procedure, especially when enlarged the experiment system, the majority RPSA complex might break down; thus we could not get the data in high

quality. (2) We also seriously considered the antibody not good enough for performing the ChIP-seq experiment in this cell line. According to the manufacturer's instructions and the reported lectures, this antibody is good enough for and mainly used in ChIP-qPCR assays. Very few works showed high specific enrichment in NGS assay with this antibody.

However, this does not affect the accuracy of the main conclusions. Based on our ChIP-qPCR results in **Fig.3b**, the RPSA binds to the promoters of proinflammatory cytokine genes like *Il6* and *Il1b*. In this system, we used the promoter of the *Ifnb* gene as the rigorous negative control to indicate the selectivity. It consisted with our molecular biology experiments, such as the co-IP experiment shown in **Fig. 4a**, HSV-1 infection enhanced the interaction between the RPSA, SMARCA and P65. We also showed IRF3 was excluded from the complex, indicating the selectivity of RPSA/SMARCA5/P65 in regulation gene expression. Notably, the co-IP experiment in **Fig.5d** also displayed that the loss of SMARCA 5 robustly impaired the interaction between RPSA and P65. The SMARCA5 complex was reported binding of distinct transcriptional factors and tended to locate at promoters in mouse embryonic stem cells or hematopoietic stem cells (Barisic, D. *et al.* Mammalian ISWI and SWI/SNF selectively mediate binding of distinct transcription factors. *Nature* **569**, 136-140 (2019).) to promote gene transcription (Ding Y. *et al.*, Smarca5-mediated epigenetic programming facilitates fetal HSPC development in vertebrates. *Blood* **137**, 190-202 (2021).). Our ChIP qPCR result in **Fig. 4c** also indicated that SMARCA5 located around the promoter regions of proinflammatory cytokine genes. Thus, our results strongly demonstrated that RPSA accelerates proinflammatory cytokine gene expression by recruiting SMARCA5 chromosome remodeling complex to gene promoters.

2). *With respect to the text near the call-out to Fig. 3c/d, the authors claim that Rpsa-iKO have less DNA accessibility near TSS regions, but Fig.3c does not seem to convincingly indicate this – at least it is not clear looking at the figure. Also, it is very*

odd that DNA promoter accessibility is shown to decrease in both WT and Rpsa-iKO samples during HSV infection even though the promoters of the genes shown are dramatically up-regulated (Fig. 3d). These observations are extra confusing because it seems that the Rpsa-iKO samples show MORE DNase accessibility on pro-inflammatory promoters, where the model proposed would suggest there should be less. I would highly recommend expanding the genome-wide profiling to include ATAC-seq at both control and infected conditions to include a more comprehensive analysis to make this data more robust instead of relying on the candidate promoter DNase hypersensitivity data.

Response: Thanks for the concerns and helpful suggestions. To make a clear view of the results, we showed all of the ATAC-seq signals and the global cluster results in both HSV-1 infected and uninfected RAW264.7 cells in **Revised Fig.3c and d**. These results showed HSV-1 infection did enhance gene transcription and loss of RPSA significantly reduced the transcriptional activity. The global clustering analysis results showed that deletion of RPSA robustly reduced a set of classical pro-inflammatory cytokine genes expression (**Revised Fig. 3d**), and the KEGG pathway analysis displayed the differential peaks from ATAC-seq mainly enriched in inflammatory responses, especially on NF-kappa B signaling pathway (**Revised Supplementary Fig. 5b**).

In **Fig.3e** in the revised manuscript, we determined the chromosome openness using DNase I sensitivity assay and calculated the results with 2^{-dCT} method relative to those uninfected samples. Thus the decreased fold change means these chromatin regions were sensitive to DNase I digestion and had high transcription activity. Our results in **Revised Fig.3e** indicated that the deletion of RPSA selectively reduced the transcriptional activity near pro-inflammatory genes promoters without affecting the *Ifnb* gene. Sorry for confusing the reviewer. We described the method and experimental procedures of DNase I sensitivity assay in more detail in the **Revised Materials and Methods**.

3) A more detailed description of the RNA-seq data is needed. For example, Supp. Fig 3b is hard to process as it contains a mix of up- and down- regulated genes relative to each of the experimental groups mentioned. I might include a global clustering/heatmap of the experiment (like shown in Fig. 2e) that provides a birds-eye-view of the gene expression changes that occur in the experiment to provide additional context. For example, it seems clear that many proinflammatory genes have reduced induction when Rpsa is depleted, but are there also a significant number of genes that are perhaps hyper-induced too?

Response: Sorry for the description not clear. We revised the descriptions of RNA-seq data rigorously in the new version. Our previous data showed the statistics of differentially expressed gene number (**Revised Supplementary Fig. 3c**) and KEGG analysis (**Revised Supplementary Fig. 3d**) of downregulated genes, but not a mixture of up- and down-regulated genes. The global cluster analysis is shown in the **Revised Supplementary Fig. 3b**. Indeed, the loss of RPSA also induced a set of genes expression. As a multifaceted protein, the RPSA has functions across various biological processes, including being the receptor of laminin and a subunit of the ribosome. Thus, the loss of RPSA may aspect functions besides interrupting host sensing pathogenic nucleic acids.

4) In Figs. 1 & 2 the authors make the claim that RPSA is a nuclear sensor for viral RNA/DNA. While several insightful experiments were performed (i.e. nuclear transfection of dsDNA), additional controls should also be considered. For example, is RPSA functional during the innate response to non-nuclear viruses (e.g. flaviviruses) or other pattern recognition receptor signals/NFkB signaling pathways (e.g. Tlr4 pathway activation/TNF)? These would appear to be important controls. Fig. 2g-h help address this, but additional stimuli would help make this more convincing. In addition, given the model that RPSA is needed to help increase pro-inflammatory gene expression during these responses, what is compensating its role in driving inflammatory gene expression in the case of cytoplasmic activation by e.g. polyIC

where RPSA is presumably not activated?

Response: We determined the phenotype in the RPSA inducible knockout RAW264.7 cells under the TNF- α stimuli in **Revised Supplementary Fig. 3f**. Our results showed that the RPSA deficiency did not affect TNF- α induced expression of *Il-1b*, *Il-6* and *Il-12b*. Strikingly, we also determined the proinflammatory cytokine gene expression with nucleus-transfected poly I:C (**Revised Fig. 2g**) cytoplasm-transfected dsDNA (**Revised Fig. 2h**) in RPSA inducible knockout and control RAW264.7 cells. Moreover, the overexpression of RPSA in RAW264.7 cells did not enhance the proinflammatory cytokine gene expression in response to cGAMP stimulation (**Revised Supplementary Fig. 3g**). Collectively, the presence of pathogenic nucleic acids in host cell nucleus is necessary for activating RPSA.

5) The following statement is not well supported in the literature: “Chromatin remodeling complexes increase chromatin accessibility to maintain transcriptional activity, which are marked by tri-methylation of histone H3 at lysine 4 (H3K4me3) around TSS of target genes”. Certainly, H3K4me3 is associated with gene TSS and active transcription, but H3K4me3 is not in and of itself proof of ISWI chromatin remodeling, as many locations where chromatin remodelers are recruited (i.e. enhancers) do not necessarily have this chromatin mark. Does Fig. 3f show all genes in the genome? Would this imply there is a global decrease in H3K4me3? How does that impact the interpretation of the data?

Response: Thanks a lot for the insightful suggestion. We carefully revised the statement in the manuscript and added new references. The **Fig.3f** shows the signal with a genome-wide view. The KEGG enrichment analysis of down-regulated differential genes was shown in **Revised Supplementary Fig. 5e**, which mainly enriched in the inflammatory responses, such as the NF- κ B signaling pathway and viral infection-related pathways. Strikingly, the **Fig. 3g** displayed H3K4me3 ChIP-seq signaling around *Il1b*, *Il6*, and *Il12b* gene sites using *ifnb* as a control. Our data showed the loss of RPSA specifically reduced H3K4me3 modification of *Il-1b*, *Il-6*,

and *Il12b* without affecting the *Ifnb* gene site.

6) The ChIP-seq profile shown in Fig. 5d is a bit concerning – most ChIP-seq data for transcription factors, including previously published p65 ChIP-seq experiments, show maximal enrichment for the TF just upstream of the TSS in the open chromatin region of the promoter. These experiments seem to show a distribution of ChIP-seq signal just downstream of the TSS, more similar to that of a ChIP-seq experiment like H3K4me3 or H3K27ac.

Response: We prudently re-checked the raw data of P65 ChIP-seq and confirmed the results in Fig. 5d are correct. As shown in the **Figure for reviewer below**, the ChIP-seq results robustly enriched NF- κ B-p65 motif, as well as the well-known co-activator motifs, such as cJun and ATF3, highly suggested our data was generated from P65 ChIP-seq. Also, as compared to the results in the revised Fig. 3f, the signal of P65 ChIP-seq mainly distributed around the TSS, which is different from the distribution features of H3K4me3 ChIP-seq signals.

Name	P-value	log P-pvalue	q-value (Benjamini)
Jun-AP1(bZIP)/K562-cJun-ChIP-Seq(GSE31477)/Homer	1e-200	-4.615e+02	0.0000
Fosl2(bZIP)/3T3L1-Fosl2-ChIP-Seq(GSE56872)/Homer	1e-189	-4.372e+02	0.0000
Fra2(bZIP)/Striatum-Fra2-ChIP-Seq(GSE43429)/Homer	1e-184	-4.251e+02	0.0000
Fra1(bZIP)/BT549-Fra1-ChIP-Seq(GSE46166)/Homer	1e-173	-4.006e+02	0.0000
JunB(bZIP)/DendriticCells-Junb-ChIP-Seq(GSE36099)/Homer	1e-170	-3.925e+02	0.0000
Atf3(bZIP)/GBM-ATF3-ChIP-Seq(GSE33912)/Homer	1e-153	-3.545e+02	0.0000
BATF(bZIP)/Th17-BATF-ChIP-Seq(GSE39756)/Homer	1e-149	-3.433e+02	0.0000
AP-1(bZIP)/ThioMac-PU.1-ChIP-Seq(GSE21512)/Homer	1e-144	-3.336e+02	0.0000
NFkB-p65-Rel(RHD)/ThioMac-LPS-Expression(GSE23622)/Homer	1e-132	-3.062e+02	0.0000

Figure for reviewer. Motif enrichment of P65 ChIP-seq.

7) *The authors mention a multitude of targets from their proteomics data, but do not report the data itself. Not only does this limit the interpretability of their results, but it is also unclear exactly how the interaction with SMARCA5 fits into those results – was it by far one of the strongest interactions, or are there a multitude of other proteins that interact with RPSA with similar strength?*

Response: In this study, we identified the candidates by integrating the mass spectrometry analysis of differential bands by CoIP-MS and quantitative proteomics results. By overlapping the top 30 binding candidates from these two data, the SMARCA5 was identified as a strong candidate which has the highest score and coverage rate. We showed the overlap proteins in **Revised Supplementary Fig. 6d**.

8) *The association of RPSA with SMARCA5 is intriguing and presents the exciting possibility that RPSA is critically involved in DNA accessibility. However, one problem is that DNA accessibility (ATAC-seq), RNAPII and TF recruitment, and chromatin marks like H3K4me3 are often highly correlated with one another and transcription. As a result, it can be hard to definitely attribute specific mechanisms when looking at changes in these data types, particularly when they change in unison as e.g. a loss of ATAC-seq signal could simply mean that the transcription of the gene is down. In addition to thinking about how the nuance that this might introduce into their interpretation of the results, I might recommend the authors consider a more targeted in vitro chromatin remodeling assay to assess RPSA's role in modulating ISWI activity.*

Response: We considered RPSA to epigenetically promote proinflammatory cytokine gene expression with comprehensive reasons. Principally and importantly, the loss of RPSA showed rare affection on the classical NF- κ B signaling transduction or the nuclear translocation of P65 (**Supplementary Fig. 4a, 4b**, and **Revised Supplementary Fig. 4c**). However, it did robustly reduce P65 binding to gene promoters (**Fig. 6f**). Moreover, our quantitative proteomics results provided the clue that RPSA predominately interacts with epigenetic regulate factors (**Supplementary**

Fig. 6b). Collectively, we conclude that RPSA promotes proinflammatory cytokine gene expression through a mechanism at epigenetic level. Following the helpful suggestion, we reorganized the manuscript carefully. And thanks for the reviewer's thoughtful advice for establishing the *in vitro* system. Since the phosphorylated RPSA could promote ISWI activity, we have tried to make recombinant RPSA Y204D protein. However, getting the functional protein was extremely difficult under current technology systems as the expressed RPSA was highly integrated and self-degradation. We will try our efforts to establish this profound significance system in the future work.

9) Have the authors considered looking at these mechanisms regulate gene expression on the viral genome? Does RPSA/SMARCA5 etc. get recruited the HSV genome? Does the RPSA-iKO show strong modulation of viral gene expression for HSV or IAV?

Response: RPSA directly binds HSV-1 gDNA (**Fig. 1j and 1k**) and IAV gRNA (**Revised Fig.11**) to promote innate response. We determine the mRNA, DNA, and protein levels of HSV-1 or IAV in the multiple RPSA-deficient cells, including A549 cells (**Revised Fig. 1b and 1c**), mouse peritoneal macrophages (**Revised Supplementary Fig. 1c**) and RAW264.7 cells (**Supplementary Fig. 4a**). These results showed that the loss of RPSA did not affect viral replications, which highly suggested that the RPSA/SMARCA5 system did not directly modulate viral gene expression.

10) Another suggestion for a relatively complicated manuscript such as this is to include a schematic/model that helps summarize the findings as a final figure panel or supplementary figure.

Response: The graphical abstract was supplied in the **Resived Supplementary Fig. 8b**.

Minor comments:

1) I would consider rewording the sentence in the abstract: “nuclear short way” does not seem very concise – I recommend replacing the term with a better description.

Response: Following the suggestion, we replaced the phrase with a new description as “an intra-nuclear way” .

2) Many mentions of “viral nuclear acids” – maybe the authors mean viral nucleic acids?

Response: We corrected it.

3) Consider including MDA/IFIH1 in the 3rd sentence for more completeness.

Response: We included this sensor in the revised manuscript.

4) Although the writing in the manuscript is relatively easy to understand, there are lots of minor language problems – needs to be carefully edited before publication.

Response: We checked the grammar and polished the text in the revised manuscript.

1) Might be nice to comment on RPSA’s previously known roles and alternative functions (and maybe how the protein was originally named).

Response: We described the known functions of RPSA in the Discussion, especially on the nucleus-located RPSA.

Reviewers' Comments:

Reviewer #1:

Remarks to the Author:

The authors made a significant effort to address all issues raised. The concerns have been satisfactorily addressed, and the results improved greatly the quality of the manuscript. The findings are novel and of high interest to a wider audience. The manuscript provides sound evidence for the identified novel sensor and propose a unique mechanism for immune defense.

Reviewer #3:

Remarks to the Author:

The revised manuscript adds additional data and analysis to help strengthen the conclusions. There is a host of compelling data in this manuscript. The data clearly establishes a role for RPSA in mediating p65-driven inflammatory responses in the context of viral infection that results in viral nucleic acids in the nucleus. In this context, it's direct role as a nucleic acid sensor and the specific role described in modulating chromatin remodeling are less well supported, but certainly possible (if not plausible) given the data collected. Overall I think the study will be interesting for readers.

Potential issues/recommendations:

With respect to RPSA acting as a sensor, one important conceptual point that might have been missed in the previous review is that Fig. 1 suggests RPSA interacts broadly with nucleic acids, which presumably include host genomic DNA or RNAs. If RPSA is truly acting as a sensor, how is it discriminating host from viral molecules? Additional context in the manuscript would help.

Based on the reviewers' response in the rebuttal to performing ChIP-seq for RPSA/SMARCA5, if the antibody/assay isn't good enough for ChIP-seq, it probably isn't very trustworthy for qPCR either. The results from the ChIP for RPSA in Fig. 3b aren't terribly convincing anyway – according to the data in the figure, RPSA is recruited to each of the promoters tested (including the *Ifnb* promoter), which is at odds with the description in the text. I might remove this data and back off any claims about RPSA's recruitment to the genome given this data is probably unreliable, and at best shows weak support for the proposed conclusions. I would say the lack of genome-wide data for RPSA and SMARCA5 make it hard to differentiate the direct activities of RPSA from general modulation in transcriptional activity.

Supp. Fig 3c suggests there is a lot going on – first, it looks like RPSA has a modest impact on expression (whereas HSV infection is the dominant factor impacting expression). Second, it looks like there are many genes with expression that is actually enhanced in RPSA-iKO cells after infection (lower-right of the heatmap) – these genes appear to be ignored in the analysis and interpretation.

As was pointed out in the first review and addressed in the rebuttal, Fig. 3e is [presumably] showing the number of PCR cycles associated with the DNase accessibility assay. This should be clearly labeled to avoid confusion – right now is simply says "relative" and no easy way for the reader to understand in higher values mean more accessibility or less accessibility without digging into the methods and description of what is presented. Not even the legend contains this information – hence the confusion in the last review and likely confusion for readers in the future.

Fig. 3c,f and 5e are tricky figures – they are used to essentially state that ATAC-seq, H3K4me3, and p65 signals are globally higher at TSS in WT vs. iKO cells. Unless careful spike-in controls were used, these types of results might suggest normalization problems or other inconsistencies between samples. For this reason, it is usually safer to analyze the specific sites that are differentially enriched (or accessible) between samples (similar to what is implied in Fig. 3d).

Reviewer #4:

Remarks to the Author:

The revised manuscript provides considerable new data supporting the model proposed by the authors. It will be very important in the discussion to explain how this proposed sensing activity integrates with well-established nucleic acid sensors such as cGAS and RLRs during viral infection. The authors have addressed the major points raised by reviewer 2 in a prior round. Their new data helps to clarify some of the key points.

Point-by-point Response to Reviewers' Comments:

To Reviewer #1:

The authors made a significant effort to address all issues raised. The concerns have been satisfactorily addressed, and the results improved greatly the quality of the manuscript. The findings are novel and of high interest to a wider audience. The manuscript provides sound evidence for the identified novel sensor and propose a unique mechanism for immune defense.

Response: We appreciate the reviewer for the positive comments.

To Reviewer #3:

The revised manuscript adds additional data and analysis to help strengthen the conclusions. There is a host of compelling data in this manuscript. The data clearly establishes a role for RPSA in mediating p65-driven inflammatory responses in the context of viral infection that results in viral nucleic acids in the nucleus. In this context, it's direct role as a nucleic acid sensor and the specific role described in modulating chromatin remodeling are less well supported, but certainly possible (if not plausible) given the data collected. Overall I think the study will be interesting for readers.

Response: We appreciate the reviewers for the positive comments.

Potential issues/recommendations:

1) With respect to RPSA acting as a sensor, one important conceptual point that might have been missed in the previous review is that Fig. 1 suggests RPSA interacts broadly with nucleic acids, which presumably include host genomic DNA or RNAs. If RPSA is truly acting as a sensor, how is it discriminating host from viral molecules? Additional context in the manuscript would help.

Response: This is a BIG open question in the field of innate sensors in the past decades, which has no answer up to now. Like the action models of well-established DNA sensor

cGAS or RNA sensor RIG-I, the nucleic acid sensors are reported to recognize viral nucleic acids, not through binding the specific sequence. That is to say, there is no-sequence specific manner of recognizing viral nucleic acids. cGAS also directly recognizes genome DNA if they meet (so there is a hypothesis of physiological separation between cGAS translocated and naked genome DNA in the nucleus). Fundamentally, both cGAS and RIG-I can recognize the naked self-genome DNA and self RNA (G.R.Pathar, *et al. Structural mechanism of cGAS inhibition by the nucleosome. Nature* **587**, 668–672 (2020); M. Jiang, *et al. Self-recognition of an inducible host lncRNA by RIG-I feedback restricts innate immune response. Cell* **173**, 906-919.e13 (2018)). How to preferentially sense the viral nucleic acids by these innate sensors remains unclear. We think that RPSA, like other nucleic acid sensors, also recognizes self-nucleic acids under pathogenic conditions such as SLE. Indeed, we found that RPSA binds naked genomic DNA but not the packaged genomic DNA (**Fig. 1K**), confirming our hypothesis. Furthermore, RPSA showed low affinity to some types of DNA, such as G3-YSD and *E.coli* ssDNA, implying the structure or modifications of DNA or RNAs might also be essential for RPSA sensing the nucleic acids. In summary, it remains unanswered how these innate nucleic acid sensors are precisely prevented from recognizing self-nucleic acids in normal conditions, although several potential ways such as physical separation proposed. We discussed these issues at **Page 19, lines 384-387**.

*2) Based on the reviewers' response in the rebuttal to performing ChIP-seq for RPSA/SMARCA5, if the antibody/assay isn't good enough for ChIP-seq, it probably isn't very trustworthy for qPCR either. The results from the ChIP for RPSA in Fig. 3b aren't terribly convincing anyway – according to the data in the figure, RPSA is recruited to each of the promoters tested (including the *Ifnb* promoter), which is at odds with the description in the text. I might remove this data and back off any claims about RPSA's recruitment to the genome given this data is probably unreliable, and at best shows weak support for the proposed conclusions. I would say the lack of genome-wide*

data for RPSA and SMARCA5 make it hard to differentiate the direct activities of RPSA from general modulation in transcriptional activity.

Response: Thanks a lot for the thoughtful suggestions. We removed Figure 3b and supplementary Fig. 4e. Since our immunoprecipitation results showed that RPSA was integrated into P65 and SMARCA5 complex (**Fig. 4a and Fig. 5d**), loss of PRSA robustly destruct their interaction (**Fig. 5c**), we essentially considered RPSA was recruited to transcriptional complexes. And the current ChIP-qPCR result also supports the hypothesis. With respect to the reviewer's concern, we back off this result. After getting proper antibodies, we will address this crucial issue further.

3) Supp. Fig 3c suggests there is a lot going on – first, it looks like RPSA has a modest impact on expression (whereas HSV infection is the dominant factor impacting expression). Second, it looks like there are many genes with expression that is actually enhanced in RPSA-iKO cells after infection (lower-right of the heatmap) – these genes appear to be ignored in the analysis and interpretation.

Response: The RPSA is a multifunction protein. Except for the role of innate sensor in enhancing pro-inflammatory cytokine gene expression, we already observed that it has some other functions, like mediating cell death at a very late period during virus infections (data not shown). We will report other functions of RPSA in the future.

4) As was pointed out in the first review and addressed in the rebuttal, Fig. 3e is [presumably] showing the number of PCR cycles associated with the DNase accessibility assay. This should be clearly labeled to avoid confusion – right now is simply says “relative” and no easy way for the reader to understand in higher values mean more accessibility or less accessibility without digging into the methods and description of what is presented. Not even the legend contains this information – hence the confusion in the last review and likely confusion for readers in the future.

Response: We revised all the descriptions in the figures and figure legends (**revised Figure 3d, Revised Supplementary Fig. 5d and 5e**).

5) Fig. 3c, f and 5e are tricky figures – they are used to essentially state that ATAC-seq, H3K4me3, and p65 signals are globally higher at TSS in WT vs. iKO cells. Unless careful spike-in controls were used, these types of results might suggest normalization problems or other inconsistencies between samples. For this reason, it is usually safer to analyze the specific sites that are differentially enriched (or accessible) between samples (similar to what is implied in Fig. 3d).

Response: We showed the enrichment signal of ATAC-seq data at the *Il-1b* gene site, using *Ifnb* as the negative control (**Revised Fig.3c, Page 10, lines 194-195**). Accompanied with Fig. 3b, our results indicated RPSA predominantly facilitated the accessibility of proinflammatory cytokine genes without affecting *Ifnb* transcription. The **Revised Fig 3f** (previous Fig.3g) analyzes H3K4me3 ChIP-seq data (**Fig. 3e**) on the specific gene sites. Here, we also used *Ifnb* as the negative control. We showed the heatmap of clustering of P65 down-regulated enriched peaks caused by RPSA lost after HSV-1 infection (**Revised Supplementary Fig. 7b, Pages 15-16, lines 308-311**).

To Reviewer #4:

The revised manuscript provides considerable new data supporting the model proposed by the authors. It will be very important in the discussion to explain how this proposed sensing activity integrates with well-established nucleic acid sensors such as cGAS and RLRs during viral infection. The authors have addressed the major points raised by reviewer 2 in a prior round. Their new data helps to clarify some of the key points.

Response: Thanks a lot for the positive comments. We added new context in the Discussion (**Pages 19-20, lines 392-396**).

Reviewers' Comments:

Reviewer #3:

Remarks to the Author:

Revisions by the authors have improved the manuscript and addressed my most pressing concerns

Point-by-point Response to Reviewers' Comments:

To Reviewer #3:

Revisions by the authors have improved the manuscript and addressed my most pressing concerns.

Response: We appreciate the reviewer for the positive comments.